# A-MYB and BRDT-dependent RNA Polymerase II pause release orchestrates transcriptional regulation in mammalian meiosis

Adriana K. Alexander[1,2], Edward J. Rice[1], Jelena Lujic[2], Leah E. Simon[2], Stephanie Tanis[2], Gilad Barshad [1], Lina Zhu [1], Jyoti Lama[2], Paula E. Cohen [2,3] ✉ & Charles G. Danko [1,2,3] ✉

During meiotic prophase I, spermatocytes must balance transcriptional activation with homologous recombination and chromosome synapsis, biological processes requiring extensive changes to chromatin state. We explored the interplay between chromatin accessibility and transcription through prophase I of mammalian meiosis by measuring genome-wide patterns of chromatin accessibility, nascent transcription, and processed mRNA. We find that Pol II is loaded on chromatin and maintained in a paused state early during prophase I. In later stages, paused Pol II is released in a coordinated transcriptional burst mediated by the transcription factors A-MYB and BRDT, resulting in ~3-fold increase in transcription. Transcriptional activity is temporally and spatially segregated from key steps of meiotic recombination: double strand breaks show evidence of chromatin accessibility earlier during prophase I and at distinct loci from those undergoing transcriptional activation, despite shared chromatin marks. Our findings reveal mechanisms underlying chromatin specialization in either transcription or recombination in meiotic cells.

Spermatogenesis is a multistep process required for the daily production of millions of spermatozoa within the seminiferous epithelium of the testis[1]. Meiosis is a critical stage in spermatogenesis during which the genome is first replicated and then halved to produce four haploid gametes. The first and longest stage of meiosis is prophase I, during which dramatic changes in chromatin occur to facilitate the events of synapsis and meiotic recombination[2]. Upon entry into prophase I, the topoisomerase-like protein SPO11, together with multiple accessory proteins, catalyzes the induction of hundreds of DNA double-strand breaks (DSBs) across the genome[3]. Of these, 90% will resolve as non-crossovers, while the remaining 10% will resolve as crossovers, which result in the physical connection between two distinct chromosomes[4]. In mammals, DSB induction initiates the assembly of a tripartite protein structure called the synaptonemal complex (SC), which tethers homologous chromosomes and facilitates the

progressive repair of DSBs[5,6]. By the end of prophase I, when the SC begins to break down, the DSB-initiated crossovers tether homologous chromosomes until the first meiotic division[2,7]. These events, from DSB formation through the first meiotic division, require a complicated and intricate series of changes to chromatin that are essential for both meiotic recombination and the proper segregation of DNA into haploid gametes.

In addition to meiotic recombination, prophase I spermatocytes have an essential role in enacting complex transcriptional programs that control events during meiosis as well as the final post-meiotic differentiation stages of spermatogenesis, known as spermiogenesis[8]. Prophase I spermatocytes initiate the transcription of thousands of protein-coding genes, pseudogenes, piRNAs, long non-coding RNAs, and transposable elements, establishing a highly complex transcriptional program[8–13]. Many of the genes

[1]Baker Institute for Animal Health, College of Veterinary Medicine, Cornell University, Ithaca, NY 14853, USA. [2]Department of Biomedical Sciences, College of Veterinary Medicine, Cornell University, Ithaca, NY 14853, USA. [3]Cornell Reproductive Sciences Center (CoRe), Cornell University, Ithaca, NY 14853, USA. ✉e-mail: paula.cohen@cornell.edu; dankoc@gmail.com

transcribed during this stage are necessary to facilitate meiotic recombination and synapsis during prophase I itself[14]. In addition, several genes that are critical for spermiogenesis are expressed during prophase I, while others, such as the Y-linked *Zfy1* and *Zfy2* genes, must be transcriptionally repressed until they are needed to facilitate events during transcriptionally inert stages later in spermiogenesis[13,15]. Dysfunctional gene expression during prophase I can lead to a variety of infertility phenotypes that are characterized by either meiotic arrest or spermiogenic failure[10,16–19].

Relatively little is known about how transcription programs are controlled during prophase I. The sequence-specific transcription factor A-MYB is known to activate the transcription of protein-coding genes and piRNAs by binding to proximal and distal regulatory elements and super enhancers across the genome[19–22]. The A-MYB protein, encoded by the *Mybl1* gene, first appears in the pachytene stage of prophase I[19]. Once expressed, A-MYB transforms the regulatory environment by switching on a large number of enhancers that were inactive during earlier stages of spermatogenesis[21]. However, it remains unclear how A-MYB activates transcription during prophase I. First, we know little about how A-MYB identifies target enhancers carrying its information-poor consensus sequence, and, beyond a number of well-characterized piRNAs, the identity of its direct target genes[20]. Second, we do not know the mechanistic basis by which A-MYB activates transcription of its target genes. Finally, and perhaps most importantly, a general feature of transcriptional activation is extensive decondensation of chromatin. Currently, it remains unknown whether meiotic chromosome axis shortening, the formation of large chromatin loops, and the 3D genome reorganization of prophase I cells prohibit transcription and nucleosome positioning[23]. Therefore, a central question is how spermatocytes are able to balance the distinct, and perhaps opposing, tasks of chromatin decondensation to facilitate transcription and meiotic chromosome axis formation to aid the defining events in prophase I.

Here we integrated genomic data throughout meiosis to ask how spermatocytes achieve a balance between the hallmark events of prophase I and transcriptional activation. We performed a comprehensive analysis of gene expression and chromatin accessibility at discrete substages of meiotic prophase I: leptonema, zygonema, pachynema, and diplonema. To understand the mechanistic basis for the dynamic changes in transcriptional activity during meiotic recombination, we studied nascent transcription by mapping the location of transcriptionally engaged RNA polymerase II (Pol II) using length-extension chromatin run-on and sequencing (leChRO-seq) to avoid the stability-derived biases of steady-state mRNA sequencing[24]. Our stage-resolved maps of nascent transcription show that Pol II is loaded on chromatin and maintained in a paused state during early prophase I. Starting in pachynema, paused Pol II is released into productive elongation in a global and highly coordinated burst of transcriptional activity during which transcription is increased by ~3-fold over baseline levels. Transcriptional activation during pachynema is initiated by A-MYB, which recruits the testis-specific bromodomain protein, BRDT, to release paused Pol II into productive elongation. Both the chromatin environment and paused Pol II, which allow A-MYB binding during pachynema to activate transcription, are established by the activity of other pioneer transcription factors during earlier stages of meiosis. Finally, sites of transcriptional activity are both temporally and spatially segregated from those involved in meiotic DSB repair and recombination despite a largely shared epigenetic environment characterized by high H3K4me3 levels, allowing cells to activate transcription of essential genes and undergo DSB repair at distinct genomic loci. Taken together, these findings reveal a separation of chromatin domains, in which distinct loci are programmed during early prophase I to focus on either meiotic recombination or transcriptional activation.

## Results

### Transcriptionally active RNA Pol II is enriched in pachynema

Prophase I can be divided into 5 substages defined by the status of the SC: formation of the SC begins in leptonema, becomes progressively connected in zygonema, fully synapsed in pachynema, and disassembles during diplonema and diakinesis[2]. Meiotic recombination occurs alongside these events, with DSBs initiated in leptonema, homolog pairing, DSB repair and the appearance of non-crossovers in zygonema, and crossovers arising in pachynema. To investigate the pattern of transcriptional activity through successive substages of prophase I, we measured the abundance of RNA polymerase II (Pol II) on chromatin by immunofluorescence on chromosome spreads using antibodies that recognize the N-terminal region of the largest Pol II subunit, RPB1 (Fig. 1a–c). Immunofluorescent imaging of prophase I substages showed significant enrichment (>3-fold-increase) of chromatin-associated Pol II in pachynema and diplonema compared to leptonema and zygonema ($n_{mice} = 3$; $n_{cells} = 606$; ****$p$-value <0.0001; Fig. 1a, d). Pachynema and diplonema showed diffuse Pol II signal throughout the nucleus, except in regions of pericentromeric heterochromatin (Supplementary Fig. 1a, arrowheads). Pol II signal was also markedly reduced within the sex body (SB), the heterochromatin-rich subdomain of the nucleus in which the XY bivalent resides, in pachynema and diplonema[25]. These data show that chromatin-associated Pol II increases from zygonema to pachynema, predominantly in euchromatic regions on the autosomes.

To connect chromatin-associated Pol II with transcriptional output, we next examined the transcriptional state of the Pol II holoenzyme during each stage of prophase I. Different stages of the Pol II transcription cycle are associated with differences in the phosphorylation states of the C-terminal domain (CTD) of RPB1[26]. We performed immunofluorescence of chromosome spreads using antibodies recognizing phosphorylation of Ser5 (Ser5P), which is associated with Pol II in a promoter-proximal paused state, and Ser2P, which marks the elongating form of Pol II[26] (Fig. 1b, c, e, f). There was no significant difference in Ser5P signal intensity across prophase I substages even though total Pol II was detected at low intensity in leptonema and zygonema ($n_{mice} = 3$; $n_{cells} = 586$; $p$-value > 0.05; Fig. 1b, e). We found significantly higher levels of total Ser2P signal in pachynema and diplonema when compared to leptonema and zygonema ($n_{mice} = 3$; $n_{cells} = 677$; ****$p$-value < 0.0001; Fig. 1c, f). Taken together, these results suggest that the Pol II complexes observed in pachynema and diplonema are primarily transcriptionally active, whereas those in leptonema and zygonema are enriched for a paused transcriptional state.

To measure the amount of transcriptionally engaged RNA polymerase (Pol) in a more quantitative manner, we next used nuclear run-on assays to analyze each stage of prophase I and early stages during spermiogenesis. Nuclear run-ons use the activity of transcriptionally engaged Pol I-III to incorporate radioactive [α32P]CTP, which is measured using a scintillation counter. Notably, this approach has two advantages compared with immunofluorescence: first, it directly measures transcriptionally engaged RNA polymerases, and second it provides a quantitative readout of RNA polymerase abundance that is linear over many orders of magnitude[27,28]. Transcriptional activity was nearly three times greater (2.7-fold increase) in pachynema than in leptonema/zygonema, diplonema, and round spermatids ($n = 21$; **$p$-value = 0.0029; Fig. 1g). In contrast to immunofluorescence measurements, radioactive counts were not significantly higher in diplonema compared with leptonema/zygonema, which may be explained either by chromatin-bound RNA polymerase that was not transcriptionally engaged, substantially decreased Pol I or III in diplonema, or technical differences between experiments. Taken together, our results demonstrate the existence of a sharp burst of transcriptional activity that peaks during pachynema, perhaps by releasing paused Pol II into an active state.

## Active transcription in pachynema is driven by pause release

To determine how the robust burst of transcriptional activation observed during prophase I affects the status of transcription and chromatin on individual genes, we next measured genome-wide patterns of transcription and chromatin accessibility. We isolated cells representing leptotene/zygotene, pachytene, and diplotene stages of prophase I using either STA-PUT or FACS (see "Methods"). Staining against proteins that are defining markers of different prophase I substages (SYCP3 and γH2AX) supported a high purity in each fraction (83.5%, 77.5%, and 87%, for leptonema/zygonema, pachynema, and diplonema, respectively), with the majority of contamination representing elongated spermatids in all fractions. We analyzed each cell stage using three genome-wide molecular assays that measure different layers of gene regulation (Fig. 2a): ATAC-seq to measure chromatin accessibility, leChRO-seq to map the location and orientation of RNA Pol I–III, and RNA-seq to measure the abundance of processed mRNA. This combination of molecular assays allowed us to identify open

chromatin, classify promoter and enhancer elements based on the transcription of enhancer-templated RNAs (eRNAs)[29], and measure gene expression changes at the level of both transcription and steady-state mRNA abundance. To adjust for global changes in transcription observed during prophase I, we used radioactive nuclear run-on assays to normalize leChRO-seq signal. After sequencing 2–4 replicates from each molecular assay to a depth of 16–87 million uniquely mapped reads, we verified that the majority of variation in each sequencing library represented the stage during prophase I (PC1: 77%, 93%, and 95% for leChRO-seq, RNA-seq, and ATAC-seq, respectively) (Fig. 2b).

Immunofluorescence showed an accumulation of Ser5P throughout prophase I (Fig. 1c), which may reflect Pol II in a promoter-proximal paused state. To confirm this finding, we first focused our analysis on leChRO-seq and asked whether Pol II accumulated near the promoter in a promoter-proximal pause position, classically ~20–60 bp downstream of the transcription start site (TSS). Consistent with this model, examination of leChRO-seq data revealed an

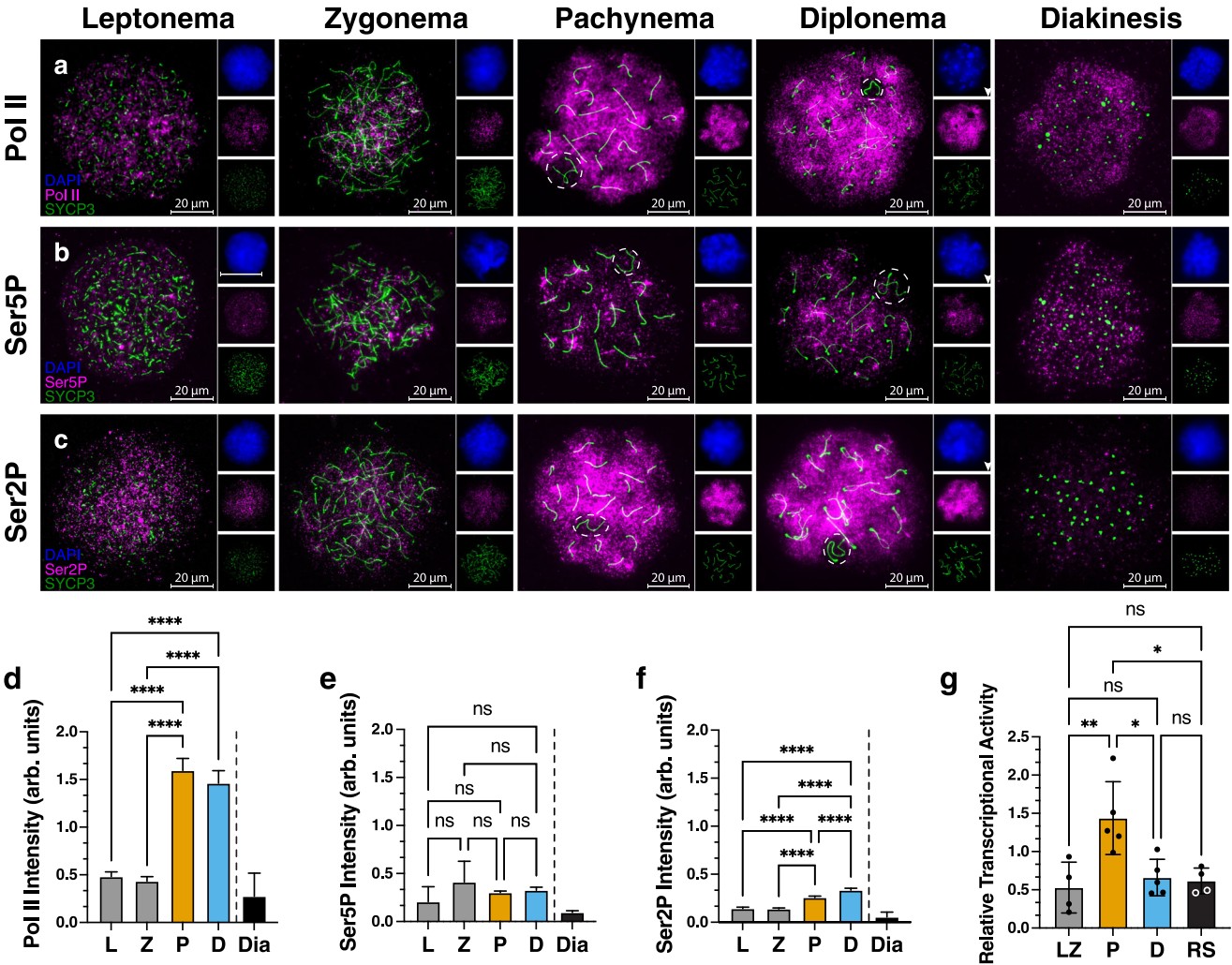

**Fig. 1 | Transcriptionally active isoform of RNA polymerase II, Ser2P, is enriched in pachynema and diplonema. a–c** Immunofluorescence staining on spread meiotic spermatocyte preparations from wild-type C57Bl/6 mice using antibodies against SYCP3 (green) and Pol II (**a**), Ser5P (**b**), and Ser2P (**c**) (magenta). Leptonema (L), zygonema (Z), pachynema (P), diplonema (D), and diakinesis (Dia) staged spermatocytes are defined by SYCP3 morphology. The X and Y chromosomes are outlined in white. Arb. units = arbitrary units. **d–f** Quantification of signal intensity for Pol II, Ser5P, and Ser2P immunofluorescence staining in wild-type spermatocytes. Average intensity is shown, and error bars represent 95% confidence intervals. Samples were obtained

from three 3-month-old mice. Source data are provided as a Source data file. **d** n = 606 prophase I cells. **e** There was no significant difference in the fluorescence intensity of Ser5P between prophase I substages. n = 586 prophase I cells. **f** n = 677 prophase I cells. **g** Radioactive nuclear run-on results comparing the relative levels of nascent transcription between leptonema/zygonema (LZ), pachynema (P), diplonema (D), and round spermatids (RS). Average relative transcriptional activity is shown, and error bars represent standard deviation. Source data are provided as a Source data file. **d–g** *p-value = 0.012, **p-value = 0.0056, ****p-value <0.0001, one-way ANOVA with the post hoc Tukey's multiple comparison's test.

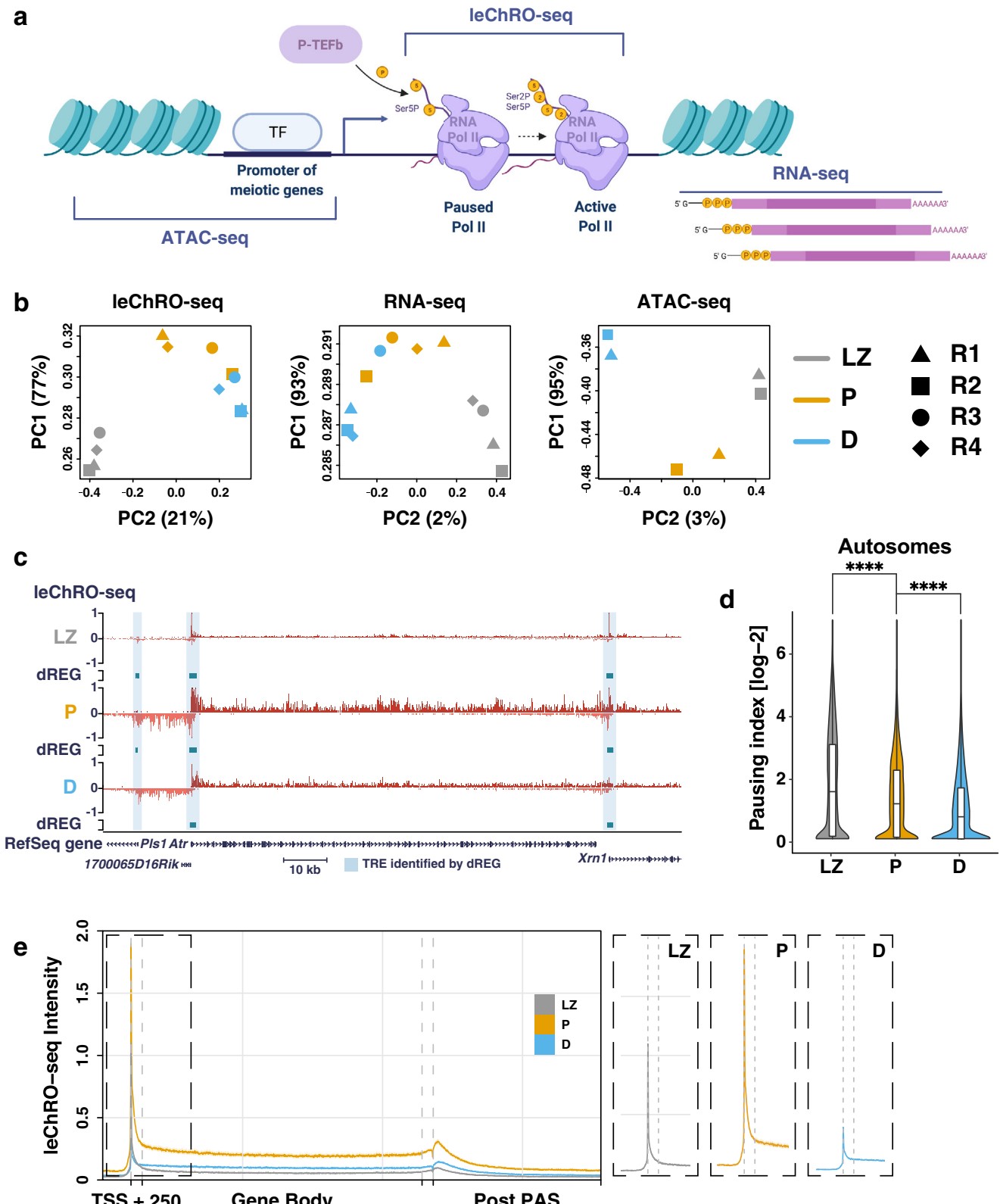

accumulation of promoter-proximal Pol II in leptonema/zygonema. For example, *Atr*, *Xrn1*, and *1700065D16Rik* all had high levels of Pol II within 60 bp of their TSS in leptonema/zygonema (*Atr*: Ataxia telangiectasia and Rad3 related, required for homologous recombination and synapsis[30]; *Xrn1*: 5'−3' exoribonuclease 1, involved in replication-dependent histone mRNA degradation[31]; *1700065D16Rik*: uncharacterized protein, expression restricted to the testis[32]). Progression

through pachynema and diplonema was associated with increased Pol II in the gene-body of all three genes (Fig. 2c).

To quantitatively examine the pattern of paused Pol II genome-wide, we computed the pausing index, or the density of paused Pol II signal relative to the signal across the gene body, which can be interpreted as the rate of pause release[33], during each stage of spermatogenesis (Fig. 2d, e). Pausing index on the autosomes and sex

**Fig. 2 | leChRO-seq detects nascent transcription in prophase I spermatocytes. a** Schematic of key steps in the transcription cycle. The transcription start site (TSS) is labeled with an arrow; nucleosomes are depicted in teal; nascent RNA is shown in dark red; and mRNA is shaded in dark pink. ATAC-seq detects accessible regulatory elements, such as promoters and enhancers. leChRO-seq provides base pair resolution of engaged and active Pol II across the genome. RNA-seq profiles polyadenylated mRNA transcripts. Created with BioRender.com. **b** Principal component analysis (PCA) plots showing variation between replicate sets of prophase I substages for leChRO-seq, RNA-seq, and ATAC-seq libraries. **c** leChRO-seq signal at the *Atr* locus for LZ (top), P (center), and D (bottom). dREG peaks (teal) are shown for each prophase I substage. TSSs overlapping with dREG peak calls are shaded in blue. Positive values represent the plus strand and negative values represent the minus strand. **d** Violin plots of pause index values calculated from the ratio of pause peak density to gene body density of leChRO-seq reads on the autosomes. ****$p < 0.0001$, two-sided Wilcoxon matched-pairs signed rank test. Median value is indicated by the horizontal, center black line. The first and third quartile are indicated by the minima and maxima of the boxplot. $n = 4$ biologically independent samples. Source data are provided as a Source data file. **e** Left: Metagene plot of median leChRO-seq signal intensity at annotated gene boundaries. LZ: leptonema/zygonema; pachynema: P; diplonema: D. Right: Inset of leChRO-seq signal for LZ, P, and D at the TSS + 250 bp.

chromosomes was highest in leptonema/zygonema, and decreased during successive stages of prophase I (Fig. 2d; Supplementary Fig. 2a). By contrast, gene body transcription on the autosomes increases as prophase I proceeds, peaking in pachynema and dropping slightly in diplonema (Fig. 2e). We conclude that increased transcriptional activity in pachytene cells is driven by paused Pol II being released into productive elongation after first being established during leptonema/zygonema.

## Pol II pausing is enriched at meiotic and post-meiotic genes

We next analyzed which annotated protein-coding genes, pseudogenes, and non-coding RNAs showed transcriptional changes during the transition between leptonema/zygonema, pachynema, and diplonema. Differential expression analysis of leChRO-seq data identified 9062 protein-coding genes, 103 pseudogenes, and 389 long noncoding RNAs (lncRNAs) that were either up- or down-regulated across all prophase I substages ($p < 0.05$, false discovery rate (FDR), Local test, DESeq2; Fig. 3a)[34]. Differentially transcribed genes in pachynema relative to leptonema/zygonema were enriched for gene ontologies related to fertilization, positive regulation of transcription by Pol II, and regulation of actin filament polymerization (log$_2$fold change > 5.40, DESeq2; Fig. 3a). Examples of the 3477 genes down-regulated in diplonema include *Ubl4b* (required for mitochondrial function[35]), *H1fnt* (essential for silencing in spermatids[36]), and *Tssk6* (involved in γH2AX formation[37]) (log$_2$ fold change < −2.83, DESeq2; Fig. 3a). Pachytene-specific lncRNAs had reported roles in phospholipid translocation, nucleoside diphosphate phosphorylation, and alternative splicing (for example: Gm11837, 1700108J01Rik, and Malat1; Fig. 3a)[38].

To identify distinct patterns of gene expression by the changes in Pol II density observed across prophase I, we clustered DE genes based on their transcription profiles in leptonema/zygonema, pachynema and diplonema according to reads falling within the TSS ± 250 bp (to capture paused Pol II), gene body, and after the polyadenylation cleavage site (PAS) (Fig. 3b, c). We identified two major clusters of genes exhibiting differences in Pol II density: *tuPAC (transcriptional upregulation in pachynema)* ($n = 1493$ genes) consisted of genes with a large increase in pachynema, which was partially, but not completely, lost during the transition between pachynema and diplonema, and *sstPAC (steady-state transcription in pachynema)* ($n = 766$ genes) which included genes that were largely unchanged in pachynema, but had decreased signal in diplonema (Fig. 3b, c, Supplementary Fig. 3a). We found that changes at the pause site through prophase I generally increased by a lower amount than the gene body for tuPAC, suggesting that the genes in this cluster are the main targets of genome-wide pause release (Fig. 3b, c). Compared to the tuPAC cluster, the genes in the sstPAC cluster did not display promoter-proximal accumulation of Pol II in leptonema/zygonema nor transcriptional bursting in pachynema. To investigate this further, we calculated the pausing index for all genes in clusters tuPAC and sstPAC for leptonema/zygonema, pachynema, and diplonema (Fig. 3d, e). Genes in tuPAC exhibited a significantly greater Pol II pausing index in leptonema/zygonema

than the genes in sstPAC (Fig. 3d; ****$p$-value < 0.00001). Moreover, pausing indices at genes in tuPAC decreased during prophase I, whereas those in sstPAC did not change. Taken together, these results suggest genes in sstPAC are transcribed in leptonema/zygonema and pachynema and regulated in a manner that is pause-independent. Conversely, tuPAC genes transcribed in pachynema are activated by the release of Pol II from a paused state that is established in leptonema/zygonema.

To determine the biological function of genes in clusters tuPAC and sstPAC, we performed a gene ontology (GO) analysis (Fig. 3f, g). tuPAC genes were enriched in biological process categories with strong relevance for transcriptional regulation (for example: mRNA processing, RNA localization, histone modification, and methylation), post-translational modification (for example: regulation of translation and regulation of protein stability), and meiosis (for example: DNA repair, chromosome segregation, and organelle fission) (Fig. 3f). We also found transcripts involved in post-meiotic processes in tuPAC, such as *Ccdc39* (involved in the motility of cilia and flagella), *Spata1* (spermatogenesis associated protein 1), and *Ybx1* and *Brdt* (testis-specific regulators of transcription). Genes in the sstPAC cluster were enriched for post-meiotic GO terms such as spermatid differentiation, sperm motility, and single fertilization (Fig. 3g). These results suggest that many of the genes involved in spermatogenesis are transcribed as early as leptonema and undergo a transcriptional decrease in diplonema. This finding may be consistent with reports that meiotic cells transcribe and store mRNAs required for the transcriptionally inert stages of spermatogenesis[13,15].

## Changes in nascent transcription lead to changes in mRNA

We next asked whether the dynamic changes in transcriptional activity across prophase I alter mRNA abundance at each stage. As we do not have a way to normalize mRNA-seq data in absolute terms (i.e., the number of transcripts per cell), we focused on examining the correlation between absolute changes in transcription and relative changes in mRNA abundance across prophase I stages. We noted reasonably strong correlations between changes at the level of transcription and mRNA abundance for DE genes (Supplementary Fig. 4a, b; $R = 0.54$–$0.63$, Pearson's correlation). Likewise, within each of the prophase I substages we observed high correlations between the leChRO-seq signal and mRNA abundance for all transcribed genes (Supplementary Fig. 4c–e; Pearson's $R = 0.78$–$0.82$). These results suggest that much of the variation in mRNA abundance during pachynema is driven by the transcriptional burst which peaks in pachynema.

Next, we asked whether mRNA stability may play a more significant role in regulating mRNA abundance as transcription shuts down between pachynema and diplonema. In diplonema, we note that the slope of the line of best fit between leChRO-seq and RNA-seq, which measures the ratio of nascent RNA transcripts to mature mRNA transcripts, decreased by 33% relative to pachynema (pachynema: Slope = 0.69; diplonema: Slope = 0.46), and the correlation between assays decreased slightly (pachynema: Pearson's $R = 0.82 \pm 0.006$; diplonema: Pearson's $R = 0.80 \pm 0.006$; Supplementary Fig. 4c–e)[39].

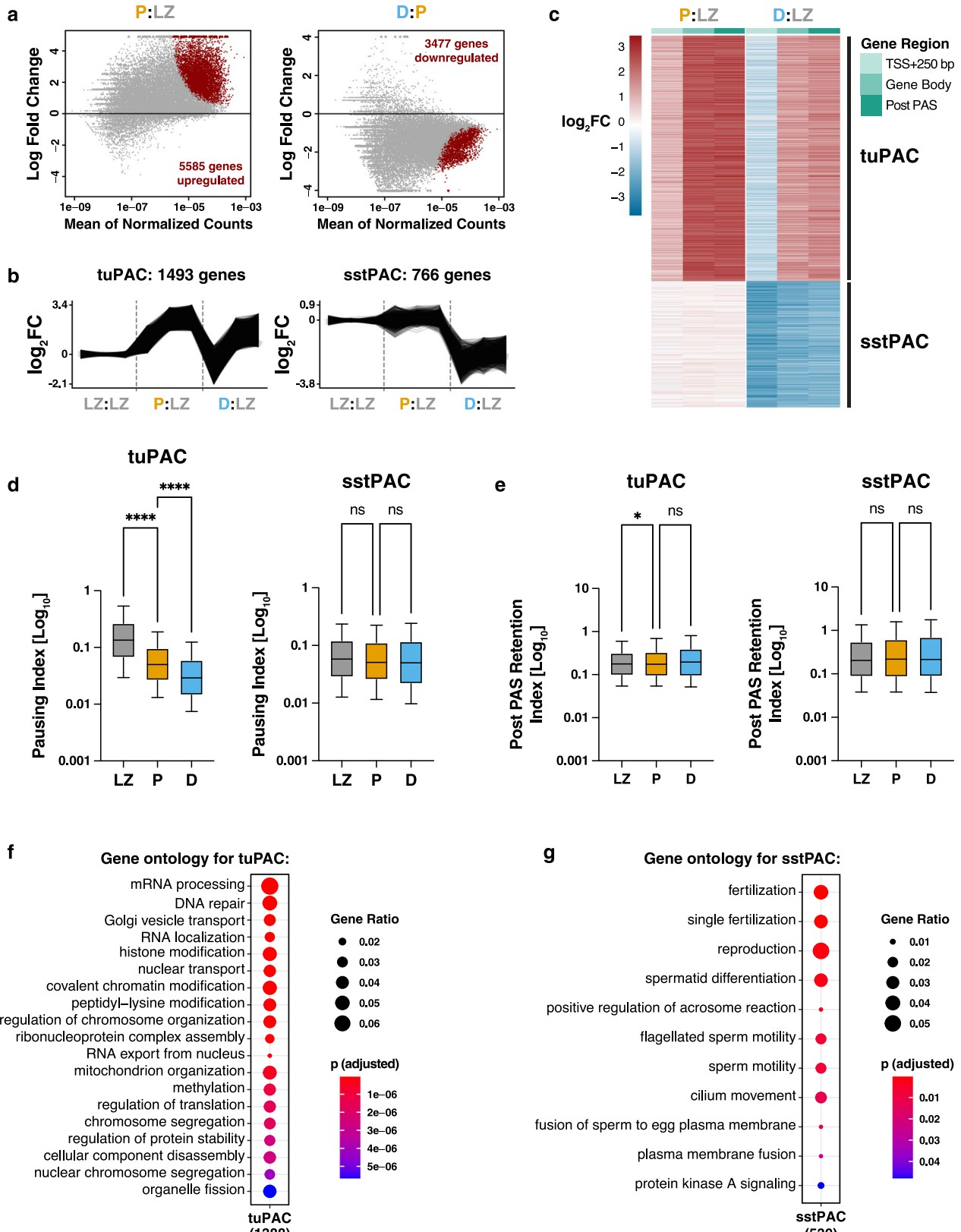

These findings may indicate that mRNA stability plays a more prominent role during later stages of prophase I as the transcriptional burst in pachynema begins to subside. To put these observations in the context of a specific genomic location, we visualized reads per million (RPM)-normalized RNA-seq libraries at the *Coa3* (Cytochrome C Oxidase Assembly Factor 3), *Cntd1* (Cyclin N-terminal Domain Containing 1),

and *Becn1* (Beclin 1; Testis Secretory Sperm-Binding Protein Li 215e) loci (Supplementary Fig. 4f). RNA-seq expression levels for these genes increased from leptonema/zygonema to diplonema, and this pattern was consistent throughout the mouse genome. Together, these observations suggest that transcription is the major determinant of mRNA abundance in leptonema/zygonema, pachynema, and

**Fig. 3 | Transcriptional changes between prophase I substages. a** MA plots showing the DEseq2-based differential expression analysis of leChRO-seq read counts within the gene bodies of all annotated genes comparing leptonema/zygonema and pachynema (left) and pachynema and diplonema (right). Differentially expressed genes are shaded in red on the MA plot. **b** Trajectories of leChRO-seq read density for individual differentially expressed genes for tuPAC (top) and sstPAC (bottom) identified in (**c**). **c** Heatmap of log$_2$-transformed fold changes of nascent RNA transcripts enriched or depleted in the selected gene regions for all differentially expressed genes in prophase I identified with DESeq2. Genes were ranked according to changes in leChRO-seq read densities at the TSS + 250 bp, gene body, and post PAS between prophase I substages. Read counts for pachynema (P) and diplonema (D) were normalized to the read counts for leptonema/zygonema (LZ). **d** Box and whisker plots of pause index values calculated from the ratio of pause peak density to gene body density of leChRO-seq reads for clusters tuPAC and sstPAC. ****$p$ < 0.0001, two-sided Wilcoxon matched-pairs signed rank test.

Median value is indicated by the horizontal, center black line. The first and third quartile are indicated by the minima and maxima of the boxplot. Whiskers represent 10–90 percentile. $n$ = 4 biologically independent samples. Source data are provided as a Source data file. **e** Box and whisker plots of post PAS retention index calculated from the ratio of the leChRO-seq density in the gene body to that in the 3′ end of the gene to 5000 bp downstream for clusters tuPAC and sstPAC. *$p$-value = 0.0176, two-sided Wilcoxon matched-pairs signed rank test. Median value is indicated by the horizontal, center black line. The first and third quartile are indicated by the minima and maxima of the boxplot. Whiskers represent 10–90 percentile. $n$ = 4 biologically independent samples. Source data are provided as a Source data file. **f, g** Gene ontology analysis for tuPAC (**f**) and sstPAC (**g**) from the heatmap in (**c**). Gene ontology processes are ranked by Benjamini–Hochberg adjusted $p$-value and the ratio of the number of genes in each gene ontology to the number of total genes in each cluster. Statistical testing was performed with an over representation analysis (ORA) using a one-sided Fisher's exact test.

diplonema, but leaves open the possibility that mRNA stability may become more important in later prophase I stages as transcription begins to shut down.

## A-MYB coordinates the pachytene transcriptional burst

We asked which transcription factors coordinate the burst of transcriptional activation in pachynema. We identified the location of promoters and enhancers, collectively called transcriptional regulatory elements (TREs), during prophase I by using dREG to identify the global initiation sites of enhancer and promoter RNAs using leChRO-seq data from each prophase I stage[29]. We identified 134,048 TREs during prophase I (Supplementary Fig. 5a), the majority of which (65%) were proximal (<1000 bp) to annotated TSSs, representing candidate promoters. To identify transcription factors responsible for transcriptional activation in pachynema, we next selected the top TREs that were differentially expressed in leptonema/zygonema compared to pachynema and searched for enriched transcription factor binding motifs (HOMER; Supplementary Fig. 5b–e)[40]. We identified 30 unique motifs with a significant FDR threshold ($p$-value <1e−54; Fisher's Exact Test), which we clustered into 5 distinct groups based on similarities in their DNA-binding specificity (Supplementary Fig. 5d, e). Similar motifs were obtained in a parallel motif discovery analysis using ATAC-seq data (Supplementary Fig. 6a), but the order was different which most likely reflects ATAC-seq marking additional types of functional elements, not all of which have a direct role in transcriptional regulation[41]. The discussion and analysis below is focused on dREG elements.

The top scoring motif bound transcription factors (TFs) in the MYB family, including *Myb*, *A-myb*, and *B-myb* (Supplementary Fig. 5e; 4a), consistent with reports that A-MYB serves as a master regulator in pachynema[19–21,42]. We next compared A-MYB ChIP-seq peaks to dREG peaks containing the binding motif for A-MYB. We found that A-MYB binding sites in pachytene spermatocytes showed a notable overlap (28%) with regulatory elements having MYB binding motifs (Chi-squared test with Yates' correction, ***$p$-value <0.0001)[20]. Collectively, our unbiased approach identified A-MYB as a master regulator of transcriptional changes in pachynema.

Next, we asked whether the genes that exhibit a transcriptional burst during pachynema (tuPAC cluster; see Fig. 3b, c) were targets of the A-MYB transcription factor. A-MYB ChIP-seq peaks were strongly enriched in genes in the tuPAC cluster (Fig. 4b)–90% of genes in the tuPAC cluster had A-MYB binding at their promoter, representing a significant enrichment relative to all active genes (****$p$ < 0.0001). Genes in the sstPAC cluster were also significantly enriched for A-MYB binding compared with all genes, although the enrichment was significantly less than observed for tuPAC genes. Furthermore, A-MYB target genes were enriched for biological functions reminiscent of tuPAC genes: mRNA metabolic processing, RNA splicing, organelle localization, and histone modification (Fig. 4c). Thus, A-MYB is a top candidate to coordinate transcriptional activation in pachynema.

## A-MYB binding correlates with BRDT binding and pause release

We next sought to determine the mechanism connecting A-MYB with the burst of transcriptional activation in pachynema. Under one model of TF-dependent pause release, a sequence-specific TF recruits transcriptional co-activators that add acetylated histone marks, which recruits the double bromodomain and extra-terminal (BET) family protein, BRD4. BRD4 connects acetylated nucleosomes with the pause-release complex, P-TEFb, to release paused Pol II into productive elongation[43]. Although existing studies have focused on somatic cells, we hypothesized a complementary mechanism exists in germ cells: namely that A-MYB acts to release paused Pol II by recruiting the testis-specific BET protein, BRDT. Therefore, we asked whether A-MYB binding is associated with the recruitment of BRDT at the tuPAC cluster of genes or the sstPAC cluster of gene promoters. We obtained BRDT ChIP-seq data from pachytene spermatocytes[44], and found that BRDT ChIP-seq peaks were 4.1-fold enriched at TSSs of tuPAC genes compared to either sstPAC or all genes (Fig. 4d; ****$p$-value <0.00001). Moreover, we found that >90% of BRDT peaks coincide with A-MYB peaks, regardless of cluster, consistent with the idea that A-MYB works in part by recruiting BRDT (Fig. 4d).

To investigate the role of BRDT in transcriptional activation in pachynema, we asked whether BRDT is localized to genes that are paused before high BRDT protein expression in pachynema. The Pol II pausing index in leptonema/zygonema was significantly greater than pachynema for genes bound by A-MYB or BRDT (Fig. 4f, g; ****$p$-value <0.00001), consistent with a model in which A-MYB and BRDT released pre-established paused Pol II upon binding in pachynema. In fact, genes that were not bound by either A-MYB or BRDT had no change in pausing index between leptonema/zygonema and pachynema (Fig. 4h), demonstrating that A-MYB and BRDT binding sites explain all of the previously observed changes in pausing index genome-wide. These results suggest a model in which A-MYB and BRDT bind near genes that have pre-established paused Pol II to facilitate pause release in pachynema (Fig. 4i, j).

## BET inhibition alters Pol II pause release in pachynema

To test our model for the involvement of BRDT in Pol II pause release in pachynema, we took a pharmacological approach to alter BRDT activity and explore pause release dynamics. We used the BET protein inhibitor thienodiazepine (+)-JQ1 (hereafter referred to as JQ1) to prevent BRDT function in prophase I cells. JQ1 inhibits the acetyl-lysine binding module of the BD1 domain of BRDT with high ligand efficiency[18]. JQ1 has been shown previously to inhibit BRDT function in spermatocytes, and this drug exhibits high testicular bioavailability without affecting hormone levels in male mice[18]. We injected 7-week-old male mice intraperitoneally (i.p.) with either JQ1 or vehicle solutions daily for 3 weeks (Fig. 5a). We performed immunofluorescence using antibodies recognizing Ser5P (paused Pol II) and Ser2P (elongating Pol II) to analyze meiotic chromosome spreads obtained from

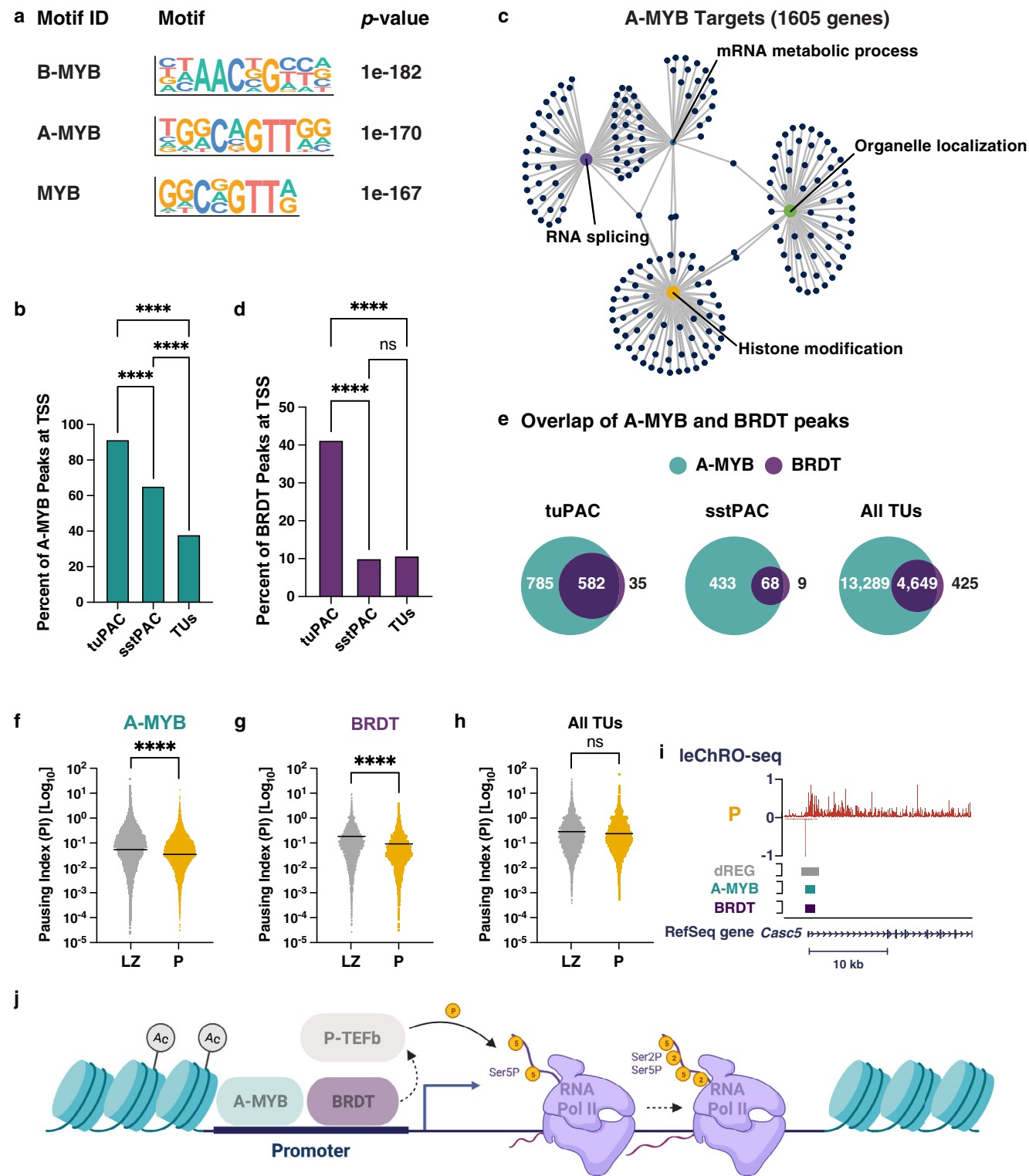

**A-MYB-dependent recruitment of BRDT to highly paused genes in Pachynema**

mice after 3 weeks of JQ1 or vehicle control (Fig. 5b, c). We found significantly higher levels of Ser5P immunofluorescence signal in pachytene and diplotene spermatocytes from JQ1-treated mice when compared to control spermatocytes (****p-value <0.0001; Fig. 5b, d). By contrast, total Ser2P signal was significantly lower for JQ1-treated spermatocytes in pachynema and diplonema when compared to the vehicle treatment (****p-value <0.0001; Fig. 5c, e). Together, these data indicate that small-molecule inhibition of BRDT results in reduced

levels of Pol II pause release in prophase I spermatocytes, leading to an accumulation of paused Pol II and a depletion of elongating polymerase.

**A-MYB binding sites are accessible prior to A-MYB expression**
We asked whether A-MYB alters chromatin accessibility as its protein binding increases during entry into pachynema. A-MYB protein is reported to reach peak abundance in pachynema[19], consistent with

**Fig. 4 | A-MYB and BRDT co-occupy the transcription start sites of differentially expressed genes with higher levels of 5′ Pol II density. a** Transcription factor binding motif enrichment revealed that the MYB transcription factor family displayed the most significant enrichment in differentially transcribed transcriptional regulatory elements in pachytene spermatocytes (FDR < 0.01; HOMER[40]; one-sided Fisher's Exact Test). The MYB transcription factor binding motifs were identified by HOMER. **b, d** The percent of A-MYB (**b**) or BRDT (**d**) ChIP-seq peaks overlapping with the transcription start site of genes in tuPAC, sstPAC, or all transcription units. Statistical analyses comparing the percent of A-MYB or BRDT peaks overlapping with the TSSs of genes in tuPAC and sstPAC used a two-sided Fisher's exact test. Statistical analyses comparing the percent of A-MYB or BRDT peaks overlapping with the TSSs of genes in tuPAC or sstPAC and all TUs used the Chi-squared test with Yates' correction. ****$p$-value <0.00001. $n$ = 4 biologically independent samples. **c** cnetplot representing the clustering of A-MYB target genes based on gene ontology enrichment analysis. **e** Representative overlap of A-MYB or BRDT ChIP-seq peaks at genes in tuPAC, sstPAC, or all transcription units. **f–h** Violin plots of the pausing indices for genes bound by A-MYB or BRDT or not bound by A-MYB or BRDT (All TUs) calculated from the ratio of pause peak density to gene body density of leChRO-seq reads. ****$p$ < 0.0001, two-sided Wilcoxon matched-pairs signed rank test. Source data are provided as a Source data file. **i** Representative example of A-MYB and BRDT co-binding at the transcription start site of *Casc5*, a gene found in tuPAC. **j** A model for the A-MYB dependent recruitment of BRDT to a gene with high levels of 5′ Pol II density in pachynema. We posit that BRDT plays a key role in the regulated release of paused Pol II during prophase I. Created with BioRender.com.

observations of both transcription and mRNA of the A-MYB-encoding gene, *Mybl1* (Supplementary Fig. 5f). At chromatin accessible regions, ATAC-seq signal peaked in pachynema similarly to nascent transcription (Fig. 6a–d). The majority of A-MYB binding sites, however, show a strong ATAC-seq signal in leptonema/zygonema, prior to peak levels of A-MYB protein expression, and retained consistently high levels of accessible chromatin throughout prophase I (Fig. 6a, e). Thus, chromatin accessibility at A-MYB binding sites appears to be established prior to peak expression of the A-MYB protein, perhaps through the actions of pioneer transcription factors either in leptonema/zygonema or prior to entry into prophase I. Two candidate pioneer factors include CREB and RFX2, both of which were expressed during spermatogenesis, are essential for entry into meiosis and post-meiotic germ cell differentiation[45,46], and had binding motifs enriched in transcriptionally active and accessible chromatin regions in pachynema (Supplementary Figs. 5e and 6a, b, Chi-squared test with Yates' correction, ***$p$-value <0.0001). Collectively, our results highlight A-MYB as a transcription factor which binds to poised open-chromatin regions and activates transcription by recruiting BRDT and other pause-release-associated transcriptional machinery.

### DSBs occur in accessible but untranscribed genomic regions
The designation and distribution of DSB and meiotic recombination hotspots is coordinated by PRDM9[47,48], a histone methyltransferase that catalyzes the deposition of H3K4me3 and H3K36me3 near its binding sites[49–51]. PRDM9 creates a permissive chromatin environment associated with euchromatic and transcriptionally active nuclear compartments[52,53], all of which are also histone modifications associated with active transcription.

We asked whether DSB hotspots also initiate Pol II transcription. We focused on recombination and DSB hotspots defined by sequenced mouse SPO11 oligos, which provide nucleotide-resolution maps of DSB formation (Fig. 7)[49]. As expected, DSBs were located in accessible chromatin, with a signal that peaks in leptonema/zygonema and decreases during the later stages of prophase I (Fig. 7a). Also as expected, DSBs are marked by H3K4me3 in leptonema/zygonema (Fig. 7b), but lost H3K4me3 in pachynema prior to the burst of transcriptional activation observed above (Supplementary Fig. 7a). PRDM9-independent DSBs also show a reduction in chromatin accessibility levels from leptonema/zygonema to pachynema (Supplementary Fig. 7b). However, whereas promoters show a strong signal for transcription that peaks in pachynema (Fig. 7c), both PRDM9- dependent and independent DSBs show no evidence of transcriptional activity at any point during prophase I (Fig. 7d; Supplementary Fig. 7c).

Both DSB formation and active transcription require extensive changes to chromatin that, despite some shared chromatin modifications, may not always be compatible. DSBs may therefore be segregated from active promoter and enhancer regions in order to facilitate DSB formation, crossover, and repair. To test this hypothesis, we asked whether transcriptionally active genes and DSBs overlapped more or less than expected by chance after controlling for chromatin accessibility in each prophase I stage. Approximately 20% of ATAC-seq peaks

initiated transcription, as determined by overlap with dREG, during each stage of prophase I, which is roughly the same order of magnitude observed in previous studies[54,55], and a significant enrichment compared to randomly shuffling the position of ATAC-seq peaks (Empirical $p$-value = 0; Fig. 7e). By contrast, less than 1200 ATAC-seq peaks marked by SPO11 oligos initiated transcription, which is much less than expected by a random model that accounts for chromatin accessibility (Fig. 7f; Empirical $p$-value < 0.003). We also noted no significant overlap between transcription and two other DSB markers, PRDM9 and DMC1 (Supplementary Fig. 8a, b), supporting a lack of transcription at DSBs regardless of the marker. These data suggest that, even though DSBs are indeed located in regions of accessible chromatin, they are statistically less likely to occur at transcribed ATAC-seq peaks. We conclude that although DSBs and TREs share many chromatin features, they rarely overlap, perhaps allowing each regulatory region to specialize in a distinct molecular function.

## Discussion
Prophase I spermatocytes require extensive reprogramming of gene expression to support the transition into meiosis and then to accomplish the morphological transformation of haploid germ cells during spermiogenesis[8,11,13,14,21,23,52,56,57]. Although meiotic chromosomes are highly organized by the synaptonemal complex, prophase I cells maintain a diverse, complex, and tightly regulated transcriptional program[11,23,52,56]. While recent advances in high-throughput sequencing have identified the overall structure of meiotic chromosomes and transcription hub formation, very little information is available on the coordinated recruitment of the transcription machinery to meiotic chromatin during discrete substages of prophase I. Here, we used leChRO-seq to directly map the presence of transcriptionally competent RNA Polymerase II (Pol II) at single nucleotide resolution for all meiotic prophase I substages from leptonema to diplonema (Fig. 2). We found that the transcriptional activity of Pol II is nearly three times greater in pachynema than leptonema/zygonema, diplonema, and round spermatids (Figs. 1 and 2). We provide evidence that Pol II pausing during early elongation in leptonema/zygonema is a general feature of the transcription cycle in prophase I and is connected to the rapid and synchronous activation of transcription of thousands of genes in pachynema (Fig. 2). Indeed, the differentially expressed (DE) genes most strongly associated with pausing encode essential regulators of male germ cell differentiation, including those required for meiotic progression, sperm motility, and testis-specific transcription factors (Fig. 3). These observations suggest that Pol II pausing provides a mechanism to finely tune meiotic genes to distinct regulatory cues and execute stringent spatiotemporal control of transcription during prophase I.

Our study has shown that A-MYB and BRDT mediate Pol II pause-release at differentially upregulated genes during pachynema. We demonstrated that A-MYB and BRDT localization in pachynema was significantly associated with actively transcribed genes with higher Pol II pausing indices in leptonema/zygonema (Fig. 4). Given that A-MYB binding co-occurs with H3K27ac at active regulatory elements during

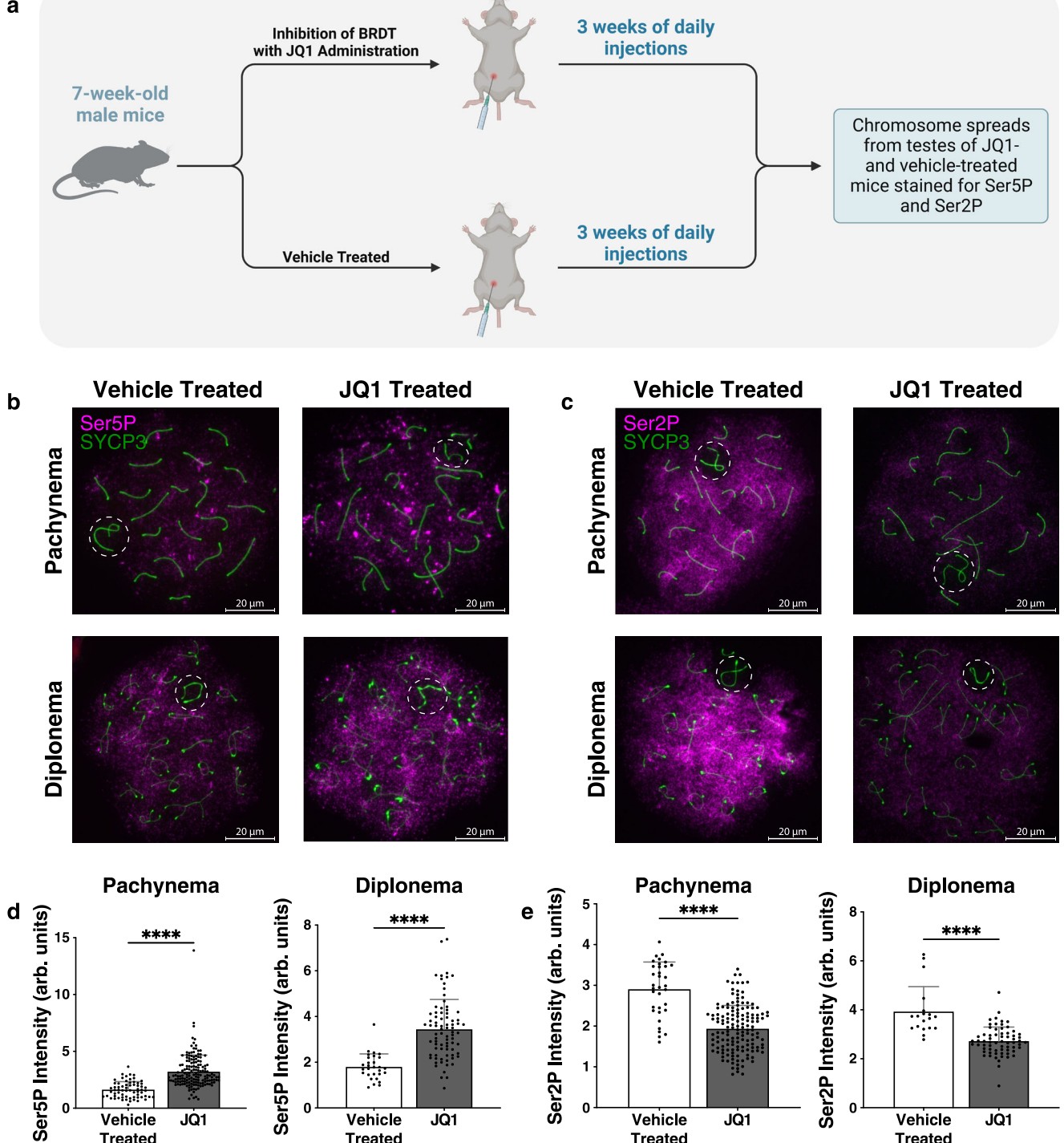

**Fig. 5 | BRDT inhibition with the small molecule, JQ1, results in altered Pol II pause-release. a** Schematic of experimental design for inhibition of BRDT with the small molecule, JQ1. 7-week-old male mice were injected intraperitoneally (i.p.) with JQ1 or vehicle solution daily for 3 weeks. Created with BioRender.com. **b**, **c** Immunolocalization of SYCP3 (green) and Ser5P (**b**) and Ser2P (**c**) (magenta) in pachytene (top) and diplotene (bottom) spermatocytes from JQ1-treated and vehicle-treated male mice. The X and Y chromosomes are enclosed within white dotted circles. **d**, **e** Quantification of signal intensity for Ser5P and Ser2P

immunofluorescence staining of pachytene (left) and diplotene (right) cells from JQ1 and vehicle-treated male mice. Average intensity is shown, and error bars represent the standard deviation. Samples were obtained from three biological replicates. Arb. units = arbitrary units. Source data are provided as a Source data file. **d** $n = 66$ vehicle-treated and 160 JQ1-treated pachytene spermatocytes; $n = 29$ vehicle-treated and 80 JQ1-treated diplotene spermatocytes. **e** $n = 34$ vehicle-treated and 148 JQ1-treated pachytene spermatocytes; $n = 19$ control and 66 JQ1-treated diplotene spermatocytes. ****$p$-value <0.0001, Mann–Whitney U test.

meiosis, we propose that BRDT is recruited to A-MYB-bound promoters through its recognition of acetylated lysine residues. Since the C-terminal domain of BRDT has been shown to interact with components of the P-TEFb complex, BRDT is likely critical for regulating Pol II

pause-release and transcriptional activation of A-MYB-bound genes during prophase I. However, we cannot exclude the possibility that other BET proteins or TFs influence the rate of pause-release in prophase I spermatocytes. Resolving the network of TFs and

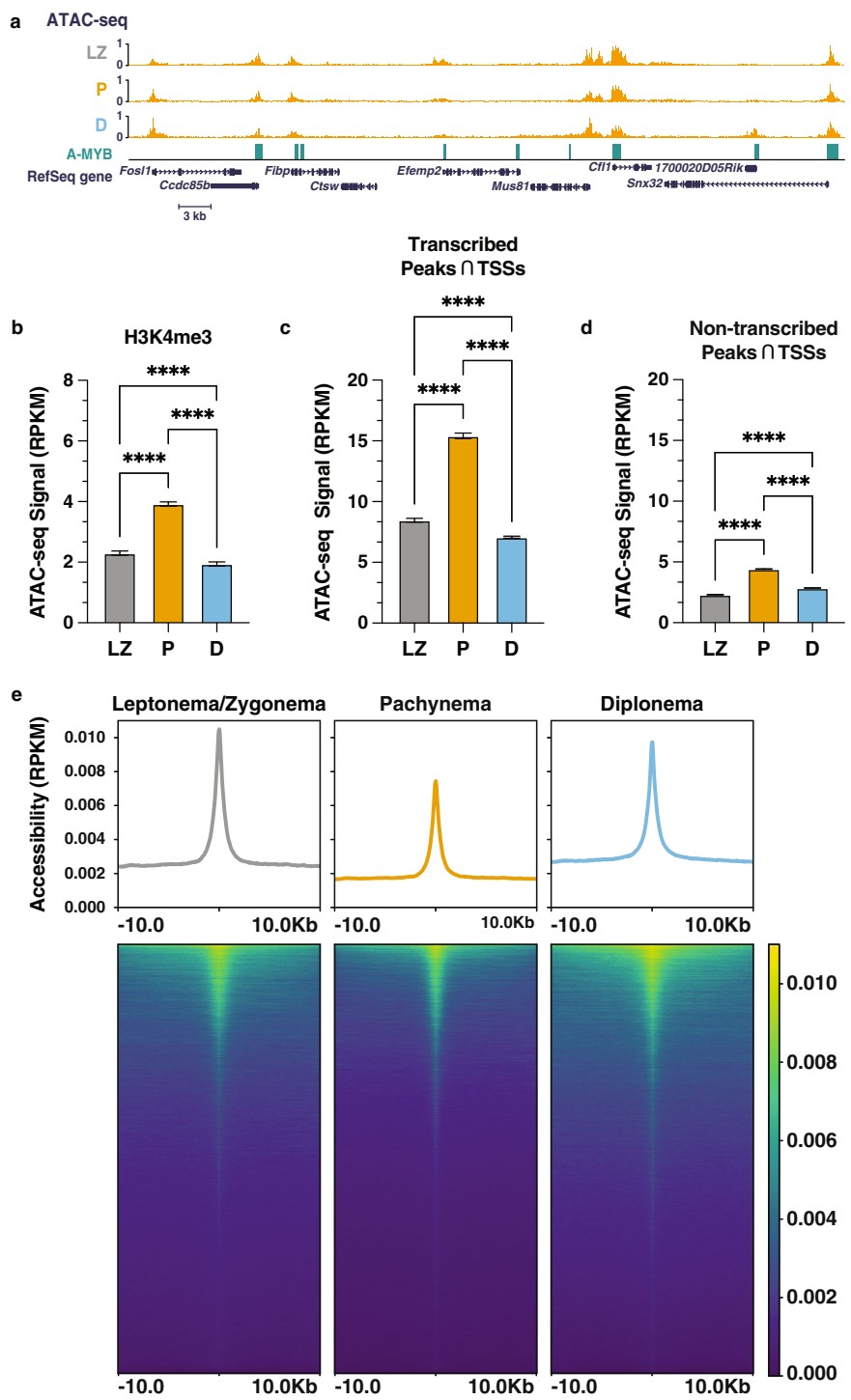

**Fig. 6 | Reprogramming of meiotic chromatin architecture is facilitated by A-MYB binding. a** Reads per kilobase million (RPKM)-normalized ATAC-seq signal at the *Mus81* locus for leptonema/zygonema (top), pachynema (center), and diplonema (bottom). A-MYB ChIP-seq peaks are shown. **b**–**d** Stage-resolved RPKM-normalized ATAC-seq signal at H3K4me3 ChIP-seq peaks (**b**); transcribed ATAC-seq peaks that overlap with annotated transcription start sites (TSSs) (**c**); non-transcribed ATAC-seq peaks that overlap with TSSs (**d**). Mean ± 95% confidence intervals are represented on bar graphs. $n = 4$ biologically independent samples. ****$p < 0.0001$, two-sided Wilcoxon matched-pairs signed rank test. Source data are provided as a Source data file. **e** Metaplots (top) and heatmaps (bottom) of RPKM-normalized ATAC-seq signal ± 10 kb from the center of A-MYB ChIP-seq peaks for leptonema/zygonema, pachynema, and diplonema. Peaks are sorted in all prophase I substages by decreasing order of ATAC-seq signal intensity in pachynema.

developmental cues responsible for Pol II pausing and transcription elongation control during prophase I provides an exciting future research direction.

One key feature of mammalian prophase I chromosomes is the spatial clustering of highly transcribed loci even as topologically associated domains (TADs), or long-range contacts, are lost in pachynema[11,23,52,56]. Thus, to determine the temporal establishment of transcriptional regulatory elements (TREs) during prophase I, we profiled active TREs at high resolution using a comparative analysis of nascent transcription and chromatin accessibility (Figs. S5 and 6).

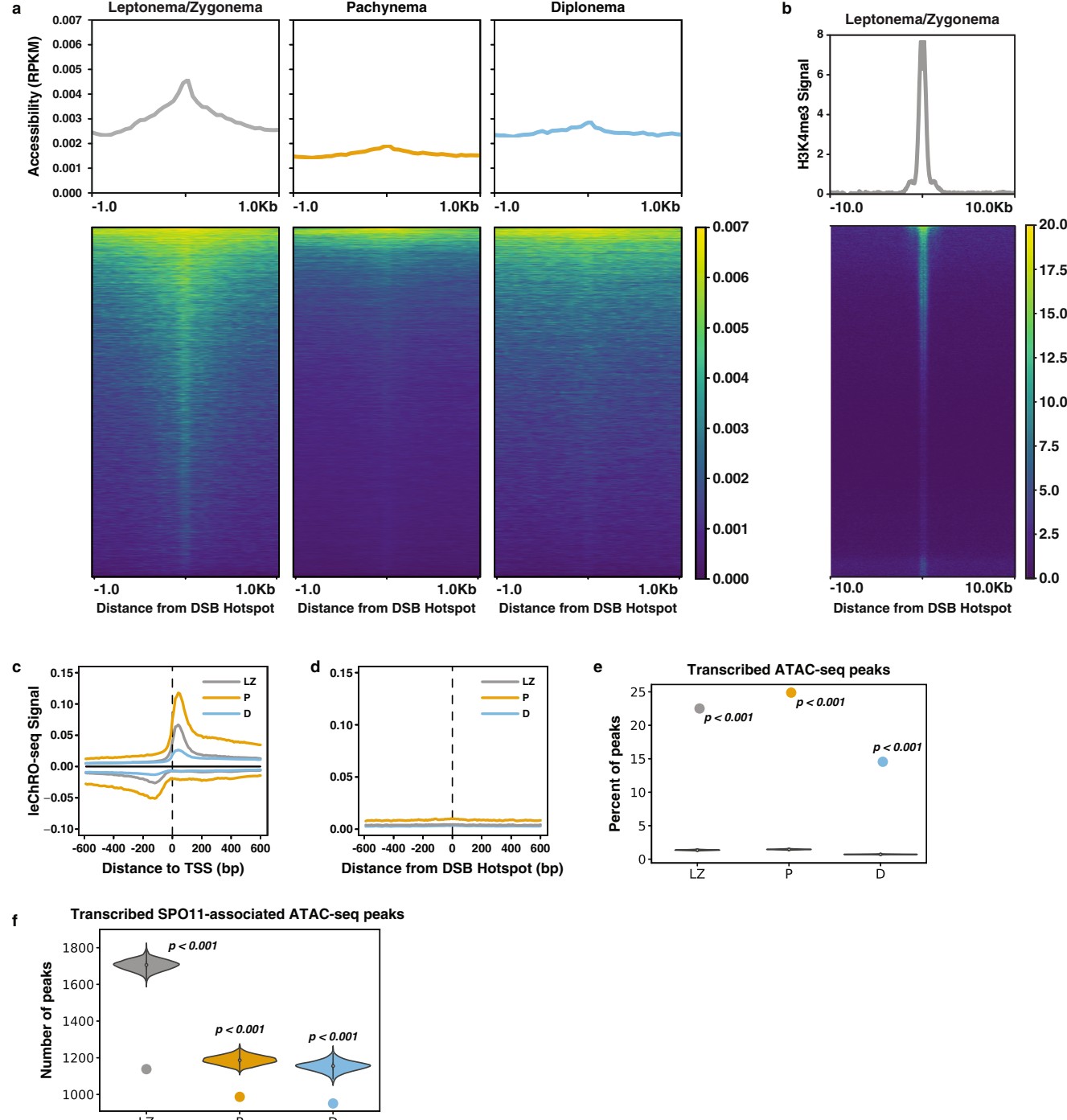

**Fig. 7 | Nascent transcription and chromatin accessibility at DSB and meiotic recombination hotspots. a** Metaplots (top) and heatmaps (bottom) of RPKM-normalized ATAC-seq signal centered on DSB hotspots inferred from SPO11 oligos for leptonema/zygonema (LZ), pachynema (P), and diplonema (D). Peaks are sorted in all prophase I substages by decreasing order of ATAC-seq signal intensity in leptonema/zygonema. **b** Metaplot (top) and heatmap (bottom) of H3K4me3 signal centered on DSB hotspots for LZ. **c, d** Metaplots of the average leChRO-seq signal centered on annotated TSSs (C) and SPO11 oligos (D) for LZ, P, and D. **e** Percentage of ATAC-seq peaks that are also transcribed based on observed overlap with dREG peaks (dot) and overlap of dREG peaks with randomly shuffled ATAC-seq peaks over 1000 iterations (violin plot). Statistical testing was performed with a one-sided Fisher's exact test; $p = 0.000999$. **f** Observed overlap of DSB hotspots inferred from SPO11 oligo data with dREG and ATAC-seq peaks (dot) and with randomly shuffled dREG and ATAC-seq peaks (violin plot). Empirical $p$-value is reported. Statistical testing was performed with a one-sided Fisher's exact test; $p = 0.000999$. Source data are provided as a Source data file.

Our data suggest that the majority of prophase I TREs are established upon entry into meiosis. Regulatory element usage in leptonema/zygonema coincides with the global accumulation of promoter-proximal Pol II and, intriguingly, paused Pol II has been shown to have a crucial role in maintaining accessible promoter chromatin architecture prior to and independently from gene activation[58]. In agreement with this observation, we found a nearly two-fold increase in genome-wide chromatin accessibility between leptonema/zygonema and pachynema at all gene TSSs (Fig. 6). Given that meiotic cells undergo profound chromosome compaction during the hallmark events of prophase I, these results suggest that chromatin accessibility is not mechanistically coupled with 3D genome reorganization at the

pachytene stage. We posit that gene-specific and local chromatin accessibility in pachynema is likely mediated by Pol II pausing, resulting in a permissive environment that enhances transcription by preventing nucleosome assembly at active promoters in pachynema.

The programmed formation of hundreds of DNA double-strand breaks (DSBs) is essential for initiating meiotic recombination and ensuring proper fertility[2]. In mice and humans, the distribution of meiotic DSBs is controlled by the binding of PRDM9 and SPO11[2,59]. To further explore the control of gene expression at DNA DSBs and recombination hotspots, we overlaid our profiles of chromatin accessibility and nascent transcription with previously published maps of PRDM9, SPO11, and DMC1 binding in C57Bl/6 males[49,53]. In particular, we found that DSB and recombination hotspots most frequently occur at regions of accessible chromatin, likely as a result of the histone methyltransferase activity of PRDM9[60]. However, engaged Pol II was undetectable at sites of meiotic recombination in pachynema, which stands in stark contrast to the widespread transcriptional activity of meiotic chromosomes at this stage. Therefore, it is conceivable that a mechanism of active transcriptional repression exists at sites of DNA DSBs and eventual crossovers or that PRDM9, SPO11, and/or DSB repair complexes actively repel the transcriptional machinery that is activated in pachynema.

In mammalian meiosis, autosomes that fail to undergo synapsis and/or recombination are subject to a surveillance mechanism termed meiotic silencing of unsynapsed chromatin (MSUC)[61,62]. MSUC deprives germ cells of transcripts via a megabase-scale chromatin remodeling process[61,62]. Meiotic silencing events are also associated with asynapsed regions of the X and Y chromosomes, which undergo synapsis and recombination only at the ~1 Mb pseudoautosomal region. However, unlike MSUC, meiotic sex chromosome inactivation (MSCI) is an obligatory event that arises as spermatocytes enter pachynema and is concurrent with the compartmentalization of the XY chromosomes into a heterochromatin-rich nuclear subdomain known as the sex body[63,64]. Disruption of MSCI induces mid-pachytene disruption of meiosis, leading to loss of germ cells and infertility[63,64]. It has remained unclear whether XY chromatin is permissive to Pol II recruitment and initiation at select loci. Here, we note that the sex chromosomes show a similar reduction in Pol II pausing index to that observed on the autosomes in pachynema, but without a concomitant increase in transcriptional activity. In the current study, the use of a highly sensitive assay to detect nascent RNA (leChRO-seq) allowed us to capture this pattern of pause release on the X and Y chromosomes, which would not be observable through bulk RNA-seq alone. The decrease in Pol II pausing indices on the sex chromosomes can be interpreted in several ways. First, assuming that A-MYB is capable of binding to XY chromatin in pachynema, induction of pause release on the sex chromosomes would be wholly possible, just as it would be on the autosomes. Second, mRNA expression of sex-linked loci may be low due to the formation of XY heterochromatin preventing productive elongation of Pol II or other mechanisms inducing rapid termination of paused Pol II on the sex chromosomes. Together, these data indicate that a reduction in Pol II pausing indices at XY genes in pachynema is not coupled with transcriptional activation as was observed on the autosomes. The precise nature of Pol II pause release regulation on the sex chromosomes will be a subject of investigation in our future studies.

In summary, we report a comprehensive profiling of Pol II occupancy genome-wide in prophase I spermatocytes. Our study directly demonstrates the global presence of paused Pol II at the TSSs of highly expressed genes in early prophase I, thus providing a mechanism for the temporal control of transcription during spermatogenesis. We examined active *cis*-regulatory elements and identified multiple TF pathways associated with the massive transcriptional activation observed in pachynema. We show that BRDT is recruited to highly paused genes in pachynema and is an important transcriptional co-activator of A-MYB-bound genes. Our observations of dynamic gene expression during prophase I provide insight into transcriptional regulation during key meiotic events, such as DSB initiation and repair. Indeed, these data represent an essential resource for the discovery of previously unknown regulatory elements and TFs coordinating the developmental transitions of spermatogenesis and ensuring successful male fertility.

## Methods

### Care and use of experimental animals
The experiments described herein used mice on the C57Bl/6J or DBA/2J background, obtained from Jackson Laboratories. All mice were housed under strictly controlled conditions of temperature (range 21–24 °F), 12 h light:day cycles, and relative humidity range from 40 to 50%, with food and water ad libitum. All mouse studies were conducted with prior approval by the Cornell Institutional Animal Care and Use Committee, under protocol 2004-0063.

### JQ1 dosage and injections
JQ1 dosage followed the original study by Matzuk et al. (2012). Briefly, JQ1 (catalog #A12729) sourced from AdooQ Bioscience (Irvine, CA) was first dissolved in DMSO at an initial concentration of 50 mg/ml and then diluted 1:10 in 10% (2-Hydroxypropyl)-β-cyclodextrin. This mixture was injected intraperitoneally (i.p.) daily into male mice at 1% of their body weight for 3 weeks for 7-week-old male mice on the DBA/2J background. Control animals received a similar i.p. daily injection of DMSO dissolved 1:10 in 10% (2-Hydroxypropyl)-β-cyclodextrin. Both control and treatment mice were weighed daily prior to injections and doses were adjusted accordingly. Three mice were used for each of the treatment and vehicle control groups.

### Chromosome spreading and immunofluorescent staining
Prophase I chromosome spreads and immunofluorescent staining were prepared as previously described[65,66]. Testis tubules were incubated in hypertonic elution buffer (30 mM Tris-HCl pH 7.2, 50 mM sucrose, 17 mM trisodium dihydrate, 5 mM EDTA, 0.5 mM DTT, 0.1 mM PMSF, pH 8.2–8.4) for 1 h. Small sections of tubules were minced in 100 mM sucrose solution and spread onto 1% Paraformaldehyde, 0.15% Triton X-100 coated slides. Slides were incubated in a humid chamber for 2.5 h at room temperature. Slides were dried for 30 min, washed in 0.4% Photoflo (KODAK, Geneva NY) diluted in PBS (800 ml in 200 mL PBS), 0.1% Triton X-100 diluted in PBS (1 mL and 199 mL PBS) and blocked in 10% antibody dilution buffer (ADB:3% bovine serum albumin, 0.05% Triton in 1 x PBS) diluted in PBS (20 mL and 180 mL). Primary antibodies used included rabbit-anti-SYCP3 (custom-made; 1:10,000 dilution)[67], mouse-anti-RNA Pol II (Millipore Sigma #05-623; 1:2000 dilution), rat-anti-RNA Pol II Ser2P (Millipore Sigma # 04-1571; 1:1000 dilution), rat-anti-RNA Pol II Ser5P (Millipore Sigma #04-1572; 1:500 dilution), and anti-A-MYB (Sigma Prestige Antibodies #HPA-008; 1:1000 dilution). Primary antibodies were diluted in antibody dilution buffer, spread across the surface of the slide, and incubated at 4 °C overnight. Slides were washed for 10 min each in 0.4% Photoflo, 0.1% Triton X, and 10% antibody dilution buffer. Alexafluor™ secondary antibodies (Molecular Probes Eugene OR, USA) were used for immunofluorescent staining at 37 °C for 1 h. Secondary antibodies were diluted in antibody dilution buffer and spread in a similar fashion to the primary antibodies. Slides were washed as previously described, dried, and mounted with Prolong Gold antifade (Molecular Probes). All secondary antibodies were raised specifically against $F_c$-fraction, $F_{ab}$-fraction purified, and conjugated to Alexafluor 488 or 594.

### Image acquisition
Images were acquired using a Zeiss Axiophot Z1 microscope attached to a cooled charge-coupled device (CCD) Black and White Camera (Zeiss McM). The images were captured and pseudo-colored by means

of ZEN 2 software (Carl Zeiss AG, Oberkochen. Germany). Exposure time was consistent between antibodies, cells, and mice. Brightness and contrast of images were adjusted using ImageJ (National Institutes of Health, USA) after fluorescence intensity measurements were calculated.

### FIJI macro script for fluorescence intensity measurements

A FIJI macro script was created using the available tools in FIJI. Images were in .czi file format, with DAPI in blue; SYCP3 in green; and RNA Pol II, RNA Pol II Ser2P, and RNA Pol II Ser5P in red. Intensities of DAPI, RNA Pol II, RNA Pol II Ser2P, and RNA Pol II Ser5P were determined using Otsu's thresholding; intensities of RNA Pol II, RNA Pol II Ser2P, and RNA Pol II Ser5P were normalized to DAPI signal intensity to account for the DNA content of each cell. The script used was as follows:

```
dir=getDir("Choose the directory containing czi files");
outfile=substring(dir,0,dir.length-1);
outfile=outfile + ".txt";
files=getFileList(dir);

run("Set Measurements…", "mean limit redirect=None
decimal=2");

//Array of averages for each channel
Dave=newArray(3);
fp=File.open(outfile);

for (k = 0; k <files.length; k++) {
        run("Bio-Formats", "open = ["+dir+files[k]+"]
color_mode=Grayscale quiet view=Hyperstack stack_
order=XYCZT");
        for (i = 0; i < 3; i ++) {
                Stack.setPosition(i+1,1,1);
                setAutoThreshold("Otsu dark");
                run("Measure");
                Dave[i]=getResult("Mean");
                }

print(fp,files[k]+"\t"+Dave[0]+"\t"+Dave[1]+"\t"+
Dave[2]);
        close();
        }

File.close(fp);
```

### Quantification and statistical analysis of images

Statistical analyses of fluorescent intensity displayed in Figs. 1 and 5 were completed using GraphPad Prism v9.0 for Macintosh (GraphPad Software, San Diego, California, USA, www.graphpad.com). Mean values were presented and alpha values were established at 0.05. All statistical analyses performed utilized One-way ANOVA with the post hoc Tukey's multiple comparison's test. Images of diakinesis-staged cells were not included in the statistical analysis due to low sample size.

### Isolation of mouse spermatogenic cells by STAPUT

Testes from adult wild-type mice (day 70–80 pp; n = 40) were extracted and decapsulated prior to enrichment of prophase I staged cell types using the gravitational cell separation method, STAPUT, which is based on separation by cell diameter and density at unit gravity[66,68]. Purity of collected fractions was determined by staining against proteins defining prophase I substages, specifically SYCP3 and γH2AX (EMD Millipore 05-636), and imaged on a Zeiss Axiophot with Zen 2.0 software. Leptotene and zygotene, pachytene, and diplotene cells were at approximately 87%, 84%, and 81% purity, respectively, with

potential contamination from spermatogenic cells of either earlier or later developmental timing. Round and elongating spermatids were identified by DAPI staining and collected at l00% enrichment. Cells isolated by STAPUT were used as input for leChRO-seq (n = 4) and RNA-seq (n = 3) library preparation.

### Isolation of mouse spermatogenic cells by FACS

Testes from adult wild-type mice (day 70–80 pp, n = 2) were extracted and decapsulated prior to enrichment of prophase I staged cell types using Flow Cytometric Analysis and Fluorescence Activated Cell Sorting (FACS)[69]. Prior to sorting, testicular single cells were stained with Hoechst dye and Propidium Iodide for selection based on DNA content and dead cell exclusion, respectively. Data analysis was done using the BD FACSDiva Software v6.1.3 on a FACSAria II cell sorter. Hoecht was excited using the 355 nm laser. Sorting flow rate was adjusted to 7–32 events/second. Supplementary Fig. 9a illustrates an example of the gating strategy used for isolating prophase I substages by FACS. Purity of collected cell populations was accessed as described above for STAPUT. Leptotene and zygotene, pachytene, and diplotene cells were at approximately 80%, 71%, and 93% enrichment, respectively. Nuclei from cells isolated by FACS were used as input for ATAC-seq libraries (n = 2).

### Chromatin isolation for nuclear run-on assays

The methods of chromatin isolation were based on work described by Chu et al., 2018[24]. For chromatin isolation from STAPUT-sorted cells, we added 1 ml of 1X NUN buffer (0.3 M NaCl, 1 M Urea, 1% NP-40, 20 mM HEPES, pH 7.5, 7.5 mM MgCl$_2$, 0.2 mM EDTA, 1 mM DTT, 50 units per ml RNase Cocktail Enzyme Mix (Ambion, 2286), 1x Protease Inhibitor Cocktail (Roche, 11 873 580 001). Samples were vigorously vortexed for 1 min. 500 ml of NUN buffer was added to each sample and vortexed for another 30 s. The samples were then incubated on ice for 30 min with a brief vortex every 10 min. Chromatin samples were centrifuged at 12,500 × g for 30 min at 4 °C. Following removal of the NUN buffer from the chromatin pellet, the chromatin pellet was washed with 1 ml 50 mM Tris-HCl, pH 7.5, with 40 units per ml RNase inhibitor, centrifuged at 10,000 × g for 5 min at 4 °C, at which point the buffer was discarded for a total of three washes. After washing, 100 ml of chromatin storage buffer (50 mM Tris-HCl, pH 8.0, 25% glycerol, 5 mM Mg(CH$_3$COO)$_2$), 0.1 mM EDTA, 5 mM DTT, 40 units per ml RNase inhibitor) was added to each sample. The samples were sonicated with a Bioruptor on high power, with a cycle time of 10 min with cycle durations of 30 s on and 30 s off. The sonication was repeated up to four times as needed to get the chromatin pellet into suspension. The samples were stored at −80 °C.

### Nuclear run-on assay using radioactive [α32P]CTP

Chromatin was isolated from STAPUT-sorted fractions containing either mixed populations of prophase I cells (n = 17) or round and elongated spermatids (n = 4). Prior to the nuclear run-on assay, DNA content of isolated chromatin was measured to ensure equal input of material per sample (500 ng of DNA). Three 10-fold serial dilutions of chromatin isolated from two Jurkat samples (Jurkat, Clone E6-1 ATCC Catalog #TIB-152) was also used to standardize each nuclear run-on experiment to account for radioactive decay of [α32P]CTP. The nuclear run-on was started by mixing isolated chromatin with 50 ml 2x chromatin run-on buffer (10 mM Tris-HCl pH 8.0, 5 mM MgCl$_2$, 1 mM DTT, 300 mM KCl, 400 μM ATP (NEB N0450S), 400 μM GTP (NEB N0450S), 400 μM UTP (NEB N0450S), 40 mM 32P-CTP (Perkin Elmer BLU008H), 0.8 units/μl SUPERase in RNase Inhibitor (Life Technologies AM2694), and 1% Sarkosyl (Fischer Scientific #AC612075000)). The run-on reaction was incubated at 37 °C for 10 min and then stopped by adding Trizol LS (Life Technologies 10296-101). RNA was extracted with 1-bromo-3-chloropropane, pelleted with GlycoBlue treated water, and washed with 100% and 75% ethanol. The RNA was

resuspended in diethyl pyrocarbonate (DEPC) $H_2O$ and any free nucleotides were captured by the use of two P-30 columns. All samples were assessed for radioactive levels on an LS Beta Counter. The relative transcriptional activity of leptonema/zygonema (LZ), pachynema (P), and diplonema (D) was calculated using the formula below:

1. *Calculate line of best fit for observed CPM of Jurkat standards*: $y = mx + b$, where $y$ = rate of ionization events per minute (CPM); and $x$ = DNA concentration
2. *Calculate expected CPM for samples using line of best fit:* substitute $x$ with measured DNA concentration of sample
3. *Relative transcriptional activity:*

$$[\text{Observed CPM} \div \text{Expected CPM}] = (\%LZ)(\mathbf{LZ}) + (\%P)(\mathbf{P}) + (\%D)(\mathbf{D}), \quad (1)$$

where **LZ**, **P**, and **D** represent the relative transcriptional activity of each stage, and were solved for using a system of linear equations

The relative transcriptional activity of round and elongated spermatids, enriched at 100%, was calculated from the *[Observed CPM ÷ Expected CPM]* ratio. Mean values ± standard deviation (SD) of the relative transcriptional activity for each cell type were presented in Fig. 1 and plotted using GraphPad Prism. Statistical analyses performed utilized One-way ANOVA with the post hoc Tukey's multiple comparison's test. Alpha values were established at 0.05.

## leChRO-seq library preparation

leChRO-seq libraries were prepared as previously described Chu et al., 2018[24]. Briefly, chromatin from enriched prophase I substages was mixed with 100 μl 2x chromatin run-on buffer (10 mM Tris-HCl pH 8.0, 5 mM $MgCl_2$, 1 mM DTT, 300 mM KCl, 400 μM ATP (NEB N0450S), 40 μM Biotin-11-CTP (Perkin Elmer NEL542001EA), 400 μM GTP (NEB N0450S), 40 μM Biotin-11-UTP (Perkin Elmer NEL543001EA), 0.8 units/μl SUPERase in RNase Inhibitor (Life Technologies AM2694), and 1% Sarkosyl (ThermoFisher #AC612075000)). The run-on reaction was incubated at 37 °C for 5 min and then stopped by adding Trizol LS (ThermoFisher 10296010). RNA was extracted from Trizol/chloroform mixture and was then pelleted with GlycoBlue and ethanol precipitation. Nascent RNA was purified by binding streptavidin beads (NEB S1421S) and washed as described by Mahat et al., 2016[70]. The RNA was removed from beads by Trizol (ThermoFisher # 15596026) and then followed with the 3′ adapter ligation with T4 RNA Ligase I (NEB M0204L). After this, a second bead binding was performed followed by a 5′ de-capping using RppH (NEB M0356S). The 5′ end of the transcript was phosphorylated using PNK (NEB M0201L) and then purified with Trizol. After this, a 5′ adapter was ligated onto the RNA transcript, and a third bead binding was performed by reverse transcription reaction in order to generate cDNA (ThermoFisher # 18090010). The cDNA was amplified (NEB M0491L) in order to generate the leChRO-seq libraries which were prepared using manufacturer's protocol (Illumina) and paired-end sequenced by means of Illumina Next-Seq500 at Cornell University Biotechnology Resource Center.

## Mapping of leChRO-seq sequencing reads

In order to align leChRO-seq data, a publically available script was used (https://github.com/Danko-Lab/utils/tree/master/proseq). The libraries were prepared by means of adapters containing a specific molecular identifier. For these specific transcripts, PCR duplicates were removed using PRINSEQ lite[71]. The adapters were cut from the 3′ end of remaining reads using cutadapt with a 10% error rate[72]. Reads from this data were mapped using BWA[73] to the mouse reference genome (mm10) in addition to a single copy of the Pol I ribosomal RNA transcription unit (GenBank ID #U13369.1). The specific location of the RNA polymerase active site was described by means of a single base which denoted the 5′ end of the nascent RNA. This information corresponded to the position of the 3′ end of each sequenced read. Mapped reads were converted to bigWig format using BedTools[74]. Libraries were normalized using the following formula:

$$For\ pachynema : (\text{bedGraph} \div \text{library size}) \times (\mathbf{P} \div \mathbf{LZ}), \quad (2)$$

where **LZ** and **P** represent the relative transcriptional activity of each stage, and (library size) × (**P** ÷ **LZ**) represents the library normalization factors.

This formula was repeated for all prophase I substages (see Table 1 for library normalization factors). The WashU Epigenome Browser was used for visualization of leChRO-seq libraries[75].

## RNA extraction and RNA-seq library preparation

Immediately following STAPUT, testicular cells were lysed with Tri-zolLS (Thermo Fisher). Total RNA was purified using TrizolLS according to the commercial protocol with the following additions: after the first phase separation, an additional chloroform extraction step of the aqueous layer was performed using Phase-lock Gel heavy tubes (Quanta Biosciences); addition of 1ul Glyco-blue (Thermo Fisher) immediately prior to isopropanol precipitation; two washes of the RNA pellet with 75% ethanol. If the RNA integrity results indicated co-purified genomic DNA, it was removed with the RapidOUT DNA Removal kit (Thermo Fisher). RNA sample quality was confirmed by spectrophotometry (Nanodrop) to determine concentration and chemical purity (A260/230 and A260/280 ratios) and with a Fragment Analyzer (Advanced Analytical) to determine RNA integrity. PolyA+ RNA was isolated with the NEBNext Poly(A) mRNA Magnetic Isolation Module (New England Biolabs). TruSeq-barcoded RNAseq libraries were generated with the NEBNext Ultra II Directional RNA Library Prep Kit (New England Biolabs). Each library was quantified with a Qubit 2.0 (dsDNA HS kit; Thermo Fisher) and the size distribution was determined with a Fragment Analyzer (Advanced Analytical) prior to pooling. Libraries were sequenced on a Next-Seq500 instrument (Illumina). At least 20 M single-end 75 bp reads were generated per library.

## Mapping of RNA-seq sequencing reads

Reads were trimmed for low quality and adaptor sequences with cutadapt v1.8; parameters: -m 50 -q 20 -a AGATCGGAAGAGCACACG TCTGAACTCCAG–match-read-wildcards. Reads were mapped to the reference genome (mm10.rRNA.fa.gz) using STAR v2.5.3a[76]. Mapped reads were converted to bigWig format using BedTools. To normalize for sequencing depth, libraries were scaled using reads per kilobase per million mapped reads (RPKM). The WashU Epigenome Browser was used for visualization of RNA-seq libraries.

## ATAC-seq library preparation

This methodology is directly based on the original design[77,78].

**Table 1 | Library normalization factors for each replicate of prophase I substage leChRO-seq libraries**

| LZ R1 | LZ R2 | LZ R3 | LZ R4 | P R1 | P R2 | P R3 | P R4 | D R1 | D R2 | D R3 | D R4 |
|---|---|---|---|---|---|---|---|---|---|---|---|
| 3176013 | 18807759 | 14145068 | 33563464 | 6470470 | 6181809 | 5708847 | 13023631 | 6697004 | 7470531 | 14690503 | 3711683 |

Library normalization values were used as input for DESeq2 size factors and for normalizing bedGraph files.

1. *Prepare nuclei:* In order to prepare nuclei, 50,000 cells isolated by FACS were spun at $500 \times g$ for 5 min immediately after sorting, followed by a wash using 50 µl of cold 1 x PBS and centrifugation at $500 \times g$ for 5 min. Cells were then lysed using a cold lysis buffer (10 mM Tris-HCl, pH 7.4, 10 mM NaCl, 3 mM $MgCl_2$, and 0.1% IPEGAL CA-630). Immediately following lysis, nuclei were spun at $500 \times g$ for 10 min in a refrigerated centrifuge. In order to avoid cell loss during nuclei preparation, a fixed-angle centrifuge was used and cells were carefully pipetted away from the pellet after centrifugation. Nuclei were stored at −80 °C.

2. *Transpose and purify.* Stored nuclei were thawed on ice and the pellet was resuspended in a transposase reaction mixture (25 µl 2 x TD buffer, 2.5 µl transposase (Illumina), and 22.5 µl nuclease-free water). The transposition reaction mixture was carried out for 30 min at 37 °C. Immediately following transposition, the sample was purified by means of a Qiagen MinElute kit.

3. *PCR.* After purification, the library of fragments was amplified using 1 x NEBnext PCR master mix and 1.25 µM of custom Nextera PCR primers 1 and 2. The following PCR conditions were utilized: 72 °C for 5 min; 98 °C for 30 s; and thermocycling at 98 °C for 10 s, 63 °C for 30 s, and 72 °C for 1 min. In order to reduce GC and size bias in the PCR analysis, PCR reactions were monitored using qPCR in order to stop amplification prior to saturation. This was done by amplification of the full libraries for five cycles, after which an aliquot of the PCR reaction was taken and 10 µl of the PCR cocktail containing Sybr Green at a final concentration of 0.6x was added. This reaction was run for 20 cycles in order to determine the additional necessary number of cycles for the remaining 45 µl reaction. The libraries were purified by means of a Qiagen PCR cleanup kit that yielded a final library concentration of ~30 nM in 20 µl. These libraries were amplified for a total of 10–12 cycles in order to generate the ATAC-seq libraries, which were prepared using the manufacturer's protocol (Illumina) and paired-end sequenced by means of Illumina NextSeq500 at Cornell University Biotechnology Resource Center.

### ATAC-seq read alignment and peak calling

ATAC-seq sequencing reads were aligned to the mouse reference genome (mm10) using Bowtie2[79] and filtered for uniquely mapping pairs with a custom python script[80]. Duplicate and multiple aligning reads were removed from the analysis with picardtools v2.1.1 "MarkDuplicates." Mapped reads were converted to bigWig format using BedTools. To normalize for sequencing depth, libraries were scaled using reads per kilobase sequence per million mapped reads (RPM). ATAC-seq peaks were called on replicate samples using Genrich v0.6 (available at https://github.com/jsh58/Genrich; parameters: -j -f all.log -q 0.05 -a 20.0 -v -e chrM -y). All peaks called by the peak calling software were included in our analysis. Mean FRiP scores for ATAC-seq libraries was 56.7%. The WashU Epigenome Browser was used for visualization of ATAC-seq libraries.

### Principle component analysis

Principle component analysis of the leChRO-seq, RNA-seq, and ATAC-seq datasets were calculated in R v3.4.2[81] using the *prcomp* function of the R *stats* package[81]. Sequencing reads from each genomics assay were counted in gene bodies of all protein-coding genes that were shown to be expressed from the leChRO-seq data and used as input for the PCA analysis. We compared the first two PCs for all datasets.

### Gene transcription analyses

**Differential expression analysis (DESeq2) of annotated genes for leChRO-seq.** We counted reads using the *bigWig* package[82] in R v3.4.2 in the intervals between (1) the annotated transcription start site (TSS) to 250 bp downstream of the annotated TSS; (2) 250 bp downstream of the annotated TSS to the 3′ end of the gene; (3) the 3′ end of the gene to

5000 bp downstream of the 3′ end of the gene. These windows were selected to count reads within the pause peak, gene body, and post poly(A) signals of all annotated genes. We limited analyses to gene annotations longer than 500 bp in length and gene annotations in which the end of the gene was at least 5000 bp away from the TSS of neighboring genes. Differential expression analysis was conducted using DESeq2[34] v1.32.0 (parameters: fit type = "local") and differentially expressed genes were defined as those with a false discovery rate (FDR) less than 0.05. Normalization of the read count matrix was achieved by manually selecting DESeq2 size factors by dividing the read count matrix by library size and multiplying by the relative transcriptional activity of the prophase I substages for each replicate (above). DESeq2 was conducted to compare differential expression between prophase I substages.

**Scaled metagene plot.** We used a publicly available script to generate leChRO metaplots (https://github.com/Danko-Lab/histone-mark-imputation/figures/ScaledMetaPlotFunctions.R). This script plots the average leChRO-seq signal intensity for all annotated genes in Fig. 2. The final plots represent a median of 1000 subsamples, without replacement. Scaled metagene plots were generated using bigWigs normalized to FPKM and relative transcriptional activity for each substage (above).

**Pausing index and post polyadenylation site (PAS) retention index.** Violin plot (Fig. 2) and box and whisker plots (Fig. 3) of the pausing index from the leChRO-seq libraries were calculated in R v3.4.2 by determining the ratio of the leChRO-seq density at the 5′ end of the gene to that in the gene body. Box and whisker plots of the post PAS retention index for the tuPAC and sstPAC clusters, presented in Fig. 3, were calculated in R v.4.0.5 by determining the ratio of the leChRO-seq density in the gene body to that in the 3′ end of the gene to 5000 bp downstream. Statistical analyses performed used the Wilcoxon matched-pairs signed rank test. Alpha values were established at 0.05.

**MA plots.** MA plots of the leChRO-seq datasets were calculated in R v 4.0.5 using the *plotMA* function of the *DESeq2* package.

**Metaplots.** Metaplots in Fig. 7 show the average signal of the sites being summarized using the *metaplot.bigWig* function of the *bigWig* R package. leChRO-seq signal was compared to publicly available mapped SPO11 oligos[49], PRDM9 binding sites[49], and DMC1 binding sites[53].

**Clustering analysis and heatmap.** The Python package *clust*[83] v1.12.0 was used to run *clust* (parameters: -n 3 -t 1.0) on the normalized read count matrix for all gene intervals of differentially expressed genes (above). The read count matrix for P and D were normalized to the read count for LZ. The *pheatmap* package[84] in R v4.0.5 was used to generate the heatmap in Fig. 3.

**dREG.** dREG was run using the default settings to identify putative transcriptional regulatory elements (TREs) from leChRO-seq libraries[29]. A complete description of dREG can be found here: https://github.com/Danko-Lab/dREG.

**Transcription factor binding site motif analysis for dREG peaks.** Briefly, we merged dREG peaks from all prophase I stages using BedTools "merge" and counted leChRO-seq reads for each prophase I substage using the *bigWig* package in R v3.4.2. Differential expression of dREG peaks comparing LZ to P and P to D was performed using DESeq2 (for normalization scheme, see above). Volcano plots reflecting DE of dREG peaks were made using the *EnhancedVolcano* package[85] v1.10.0 in R. dREG peaks with a $log_2FC > 1$ between LZ and P were selected to identify the top DE dREG peaks (28,232 total peaks) for transcription factor (TF) binding motif enrichment. HOMER[40] was used

to identify enriched TF binding motifs within upregulated dREG peaks between LZ and P, allowing multiple motifs per peak and searching for up to 25 motifs 100 bp upstream and downstream from the center of each peak. Donut plot of enriched TF families was generated using the *lessR* package[86] v4.0.5 in R. MotifScan was used to determine the genomic position of upregulated TREs in P containing the A-MYB binding motif[87]. BedTools "closest" was used to identify putative DE target genes with an FDR < 0.05 for each TRE containing the A-MYB binding motif. Putative DE target genes were defined as those nearest to each TRE containing the A-MYB motif within 50 kb.

**GO annotation and enrichment.** The GO enrichment analysis shown in Figs. 3 and 4 were carried out using the R package *clusterProfiler*[38] v4.0.5. In Fig. 3, subsets of genes identified by DE analysis carried out between prophase I substages (above) were selected based on clustering results. In Fig. 4, DEG within 50 kb of TREs were selected for the A-MYB transcription factor binding motif. These genes were used as input to the enrichGO function. The Benjamini–Hochberg method ($a = 0.05$) was applied to control for FDR. GO annotation enrichment for biological processes found in Fig. 3 were graphically represented using the *dotplot* function of the *Enrichplot*[88] v1.12.2 package. Dot plots were ranked by adjusted *p*-value and gene set size. GO annotation enrichment for biological processes found in Fig. 4 were made using the *cnetplot* function of the *Enrichplot* package.

**A-MYB and BRDT ChIP-seq data analysis**
Previously published datasets for A-MYB ChIP-seq peaks[20] (GSE44588) and BRDT ChIP-seq peaks[44] (GSE98489) were remapped from the mm9 to mm10 reference genome and mm8 to mm10 reference genome, respectively, using the UCSC LiftOver tool. Bar plots of the percent of A-MYB or BRDT peaks overlapping with the TSSs of genes in the tuPAC cluster, sstPAC cluster, or all transcription units (TUs) (Fig. 4b, d) were generated using BedTools intersect and plotted with GraphPad Prism. Statistical analyses comparing the percent of A-MYB or BRDT peaks overlapping with the TSSs of genes in tuPAC or sstPAC used Fisher's exact test. Statistical analyses comparing the percent of A-MYB or BRDT peaks overlapping with the TSSs of genes in tuPAC or sstPAC and all TUs used the Chi-squared test with Yates' correction. Bar plots showing the min, max, and median pausing index (Fig. 4f–h) from leChRO-seq libraries for genes bound by A-MYB or BRDT or not bound by A-MYB or BRDT were calculated in R v4.0.5 by determining the ratio of the leChRO-seq density at the 5′ end of the gene to that in the gene body and plotted using GraphPad Prism. Statistical analyses performed used the Wilcoxon matched-pairs signed rank test. All alpha values were established at 0.05. Metaplots and heatmaps of ATAC-seq signal centered on A-MYB ChIP-seq peaks were generated with deepTools[20,89].

**RNA-seq analysis**
Quantification of sequencing reads was performed using Salmon[90] v1.5.2 to the mm10 genome and *tximport*[91] v1.20.0 was used to summarize transcript-level to gene-level abundance estimates in R v4.0.5. Raw sequencing counts from Salmon were DESeq2 normalized (parameters: rowSums(counts(dds)) > 1). DE genes were defined as those with an FDR less than 0.05. Scatterplots comparing the $\log_2$ fold change and log FPKM of DE genes or all genes using leChRO-seq and RNA-seq data as input presented in Supplementary Fig. 4 were generated with *ggplot2*[92].

**Analysis of ATAC-seq peaks**
Peak calls from biological replicates for each prophase I substage were used to determine total unique peak sets for each substage. Peak calls representing all of prophase I were then combined and merged using BedTools. Both transcribed and non-transcribed ATAC-seq peaks were

then compared to a publicly available H3K4me3 dataset[93] and annotated TSSs using BedTools intersect. Counts of reads falling into merged peak files were determined using the BedTools multicov function for each sample, and were subsequently RPKM normalized. Mean values ± 95% confidence intervals of RPKM for each class of ATAC-seq peaks were presented in Fig. 5 and plotted using GraphPad Prism. Statistical analyses performed used the Wilcoxon matched-pairs signed rank test. Alpha values were established at 0.05. Metaplots and heatmaps of ATAC-seq signal and H3K4me3 signal centered on SPO11 oligos were generated with deepTools[49,89,93].

HOMER[40] was used to identify enriched TF binding motifs in ATAC-seq peaks, allowing multiple motifs per peak and searching for up to 25 motifs 100 bp upstream and downstream from the center of each peak. MotifScan was used to determine the genomic position of accessible regions containing the A-MYB, RFX2, and CREB binding motifs[87].

**Overlap of accessibility, transcription, and DSB markers**
We first asked whether transcription initiation, marked by dREG peaks, and the DSB markers (PRDM9 binding motifs, Spo11 oligos (GSE84689), and DMC1 ChIP-seq peaks (GSE35498)) are enriched in accessible chromatin, based on overlap with ATAC-seq peaks at the different stages of meiotic prophase I[49,53]. Using the center of the dREG or the DSB marker peaks, we asked what percentage of the ATAC-seq peaks overlap with either of these markers by counting the number of ATAC-seq peaks with values greater than zero for any of the markers following BedTools "intersect" with -C function, at all three stages. To ask if the calculated overlaps between ATAC-seq peaks and the different markers are different than what is expected by chance, we randomly shuffled the positions of ATAC-seq peaks across the genome, while preserving their size, preventing overlaps or changes in chromosome distribution with BedTools "shuffle -chrom -noOverlapping", 1000 times. In each of these 1000 iterations, we asked what percentage of the shuffled ATAC-seq peaks overlapped with the dREG and DSB markers. We then calculated an empirical *p*-value for each marker by taking the fraction of the iterations that resulted in ATAC-seq peak overlaps greater than the observed overlap.

We then asked if the accessible genomic regions presenting transcription initiation occur near or are separated from sites of DSBs. We considered a window of 5 kb around the center of ATAC-seq peaks in each of the stages and asked what fraction of these expanded peaks overlapped with each of the markers, including dREG and DSB markers. We then calculated the co-occurrence proportion of dREG and DSB markers in the same expanded ATAC-seq peak. We defined the column of dREG overlapping expanded ATAC-seq peaks as "transcribed ATAC peaks" and randomly shuffled this column 1000 times while reassessing the co-occurrence in each iteration. The empirical *p*-value was calculated by the number of iterations in which the number of ATAC-seq peaks overlapping with dREG (shuffled "transcribed ATAC peaks") and the DSB markers was smaller than the observed co-occurrence calculated.

**BioRender**
Figures 2a, 4j, and 5a were created with BioRender.com.

**Statistics and reproducibility**
No statistical method was used to predetermine sample size. No data were excluded from the analyses. The experiments were not randomized.

**Reporting summary**
Further information on research design is available in the Nature Portfolio Reporting Summary linked to this article.

## Data availability

Complete leChRO-seq, ATAC-seq, and RNA-seq data generated in this study have been deposited in the NCBI GEO database under accession code GSE212120 and are publicly available. PRDM9 motifs and Spo11 oligo data are available at the NCBI GEO database under accession code GSE84689. DMC1 ChIP-seq peak data are available at the NCBI GEO database under accession code GSE35498. Mm10 was used as the reference genome in this study and is available at the NCBI Assembly database [https://www.ncbi.nlm.nih.gov/assembly/GCF_000001635.20/]. All data presented as graphs in the Figures are provided in the Supplementary Information/Source data file. Source data are provided with this paper.

## Code availability

Custom code for this manuscript is publicly available at https://github.com/Danko-Lab/Prophase-I-Transcription-Project[94].

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

## Acknowledgements

We thank J. Lewis and C. Pereira for sharing scripts used in data analysis, J. Lis and A. Ozer for valuable discussions on radioactive run-ons, and J. Grenier and the TREx core for preparing RNA-seq libraries. Valuable discussions with A. Grimson, J. Schimenti, M. Roberson, and all members of the Danko and Cohen labs during the course of this project helped to shape our manuscript. Work in this publication was supported by R01-HG009309 (NHGRI) to C.G.D., INV-035106 and INV-003771 (Bill and Melinda Gates Foundation) to P.E.C., and P50-HD104454 (NICHD) to P.E.C. and C.G.D. The content is solely the responsibility of the authors and does not necessarily represent the official views of the US National Institutes of Health.

## Author contributions

A.K.A.: Conceptualization, methodology, formal analysis, investigation, data curation, writing-original draft, visualization. E.J.R.: Methodology, investigation, resources. J.L.: Methodology, formal analysis, investigation, visualization. L.E.S.: Methodology, formal analysis, investigation, visualization. ST: Methodology, formal analysis, investigation, visualization. G.B.: Methodology, validation, formal analysis. L.Z.: Methodology, software, validation, formal analysis. J.L.: Methodology, validation, formal analysis. P.E.C.: Conceptualization, methodology, writing-review & edit, supervision, project administration, funding acquisition. C.G.D.: Conceptualization, methodology, software, formal analysis, data curation, writing-review & edit, supervision, project administration, funding acquisition.

## Competing interests

The authors declare no competing interests.
