## [Peer Review File · Nature Communications]

A-MYB and BRDT-dependent RNA Polymerase II pause release orchestrates transcriptional regulation in mammalian meiosisREVIEWER COMMENTS

Reviewer #1 (Remarks to the Author):

The manuscript describes a detailed analysis of changes in transcription that take place during meiosis in mice. The manuscript explores the contribution of transcription initiation, promoter-proximal pausing, and RNA stability to these changes. The authors then examine the location of double-stranded breaks and their possible overlap with active enhancers and promoters. From these analyses, the authors conclude that, although DSBs are located in regions containing active histone modifications, their location is not transcribed and therefore, DSB do not overlap with transcribed genes.

The manuscript addresses an interesting and important topic. The experimental design is excellent, and the analysis of genomics data is well done. The manuscript is also logically written and easy to follow. I have no major concerns and I think the manuscript is appropriate for publication in Nat. Comm. The following are some comments the authors may want to consider:

Major Comments

1. ATAC-seq measures chromatin accessibility due to the presence of a bound transcription factor at the summit of the ATAC-seq peaks. The authors determined TFs responsible for transcription activation in pachynema by using dREG to identify eRNAs in leChRO-seq data. This seems like a less accurate method than using ATAC-seq peaks. Is A-MYB identified as the top TF when using ATAC-seq data? The authors find that the majority of A-MYB binding sites show ATAC-seq signal. Is the reverse also true, that most ATAC-seq peak summits contain A-MYB binding motifs? Do these peaks contain motifs for CREB and RFX2?

Minor comments

1. Line before last in the abstract. Perhaps "mechanisms" instead of "the mechanism" would be more appropriate in case there are other mechanisms.

2. Page 2 "RNAs transcribed during prophase I may also be stored until they are needed to facilitate events during transcriptionally inert stages later in spermiogenesis". Could these RNAs be made in round spermatids?

3. Bottom of page 2 and top of page 3 "a central question is how spermatocytes are able to balance the separate, and arguably opposing, tasks of chromatin decondensation to facilitate transcription and chromosome condensation to aid the defining events in prophase I". Authors should consider whether the level of condensation (formation of large loops?) that results in the formation of prophase chromosomes is the same as the level of condensation that interferes with transcription (nucleosome positioning?).

4. Figure 1. Perhaps it would be more intuitive to have Ser2p below Ser5p in this figure.

5. Page 10. "Thus, we conclude that gene expression profiles for leptoneuma/zygoneuma, pachynema, and diploneuma have notably different transcription programs". Please explain how this conclusion was reached. It seems from the text that it was based on just a few genes. Alternatively, tone down the statement or delete the sentence.

6. Please refrain from referring to covalent histone modifications as "markers".

7. Page 21, bottom. The authors seem to imply that DSBs do not overlap with ATAC-seq peaks. Is this correct? Please explain more explicitly in the text.

Reviewer #2 (Remarks to the Author):

In this manuscript, Alexander et al. generated time-series transcription-related omics data (mRNA-seq, leChRO-seq, and ATAC-seq) throughout the mouse meiosis prophase I to ask how spermatocytes achieve pachytene transcription burst and keep a balance between meiotic recombination and transcriptional activation through integrated analysis. The authors analyzed the average density of paused and elongated RNA polymerase in different stages through immunofluorescence staining and inferred that pachytene transcription burst is driven by released RNA pol II paused in leptotene and zygotene. Radioactive nuclear run-on and leChRO-seq data supported this in quantitative analyses. Then they identified the candidate TFs regulated the RNA pol II release via motif analysis using de novo dREG peaks found in pachytene and focused on MYBL1 and BRDT. By re-analysis of the previously reported MYBL1 and BRDT ChIP-seq data, the authors proposed a model of MYBL1-dependent recruitment of BRDT to highly paused genes in pachytene to release the paused RNA pol II. Finally, they compared transcribed regions and DSB hotspots and found these two regions are mutually exclusive on the genome despite with similar chromatin environments.

The omics data are of good quality, and the analysis was performed systematically and logically. The immunofluorescence staining of RNA pol II and leChRO-seq supports that the essence of pachytene transcriptional activation is releasing of paused Pol II. The occupancy sites of MYBL1 and BRDT showed a strong correlation with the pachytene-upregulated genes. Their targets showed a reduced pausing index in pachytene compared to leptotene/zygotene. This association leads to a much clearer model of how pachytene transcription burst works. However, I am concerned about the main conclusion that meiotic gene transcription programs are mediated by A-MYB and BRDT-dependent RNA polymerase II pause release because no data from knock-out mice were involved in this work. The leChRO-seq in *Brdt*^{-/-} pachytene spermatocytes may provide direct evidence for this hypothesis model.

Specific comments:

1. In Figure 1, I noticed distinct signals of Ser5P near the sex chromosomes (1C), where no Pol II (1A) and Ser2P (1B) signal was detected. This indicates Pol II is paused at sex chromosomes which is opposite to autosomes in pachytene and diplotene. It is quite interesting to discuss this phenomenon from the perspective of Pol-II.
2. The sex chromosome should be treated separately when performing transcription-related omic analysis due to MSCI. Are there any peaks of leChRO-seq at chrX in pachytene and diplotene? If so, how does the pausing index of chrX genes change at pachytene and diplotene compared to leptotene/zygotene?
3. I am confused about the negative values in leChRO-seq track in Fig 2C & Fig S3F. Authors should make a clearer description in figure legend.
4. Some figure legends are not matched to corresponding figures (e.g. Fig S2, Fig. 4).
5. In "A-MYB coordinates the pachytene transcriptional burst" part, the description of Fig S3A is not consistent with Fig. S3A
6. In Fig 3B & C, the global cluster pattern should be provided.
7. The description of "A-MYB ChIP -seq data from pachytene spermatocytes showed a notable overlap (28%) with regulatory elements having MYB binding motifs" is quite confusing. Do authors compare A-MYB binding sites versus dREG with MYB motifs?
8. In Fig. 5C & D, the union symbol may be an intersect symbol according to the description in the figure legends.
9. In Fig 6E, why only 5% of dREGs are overlapped with the ATAC-seq peaks, considering that transcribed regions should be accessible.
10. In Fig S5, why overlapped regions of PRDM9 are less than DMC1, considering that more PRDM9 occupancy sites are observed than DMC1?
11. In methods, more information about which genes were used for PCA. All genes or genes with filtration?

12. If loss of BRDT, the genes with a reduced pausing index in pachytene compared to leptotene/zygotene may have a tendency to be downregulated. The RNA-seq data of 20 dpp Brdt^{-/-} testis in GEO database under GSE39909 may be helpful.

Reviewer #3 (Remarks to the Author):

In this work, Alexander and collaborators report a genomic profiling of Pol II occupancy in mouse primary spermatocytes. To do so, the authors combine leChRO-sequencing with RNA-seq and ATAC-seq in flow sorted enriched populations, distinguishing between leptotema/zygonema, pachynema and diplotema spermatocytes. First the authors report an increased transcriptional activity in pachytne cells, suggesting that this is driven by the release of Pol II that has been previously paused in leptotema/zygonema. Following this finding, the authors seek to find which transcriptional factors might be driving this phenomenon, finding a high correlation to A-MYB binding sites. Not surprisingly, the authors do not find correlation between the location of DSBs and transcription.

The paper is well written, and I enjoyed going through it. I find the authors provide compelling evidence of the presence of paused Pol II at TSSs of highly expressed genes in early prophase I. I have though a couple of comments that might need further clarification.

First, it will be helpful to provide a more detailed description of the isolation of mouse spermatocytes by FACS. This can be added to the supplementary material, including a more detailed description of the methodology and flow cytometry plots showing different populations isolated. I am bringing this up because it seems that in the PCA plots from Fig 2, the PC2 is not able to set completely apart pachytene from diplotene cells (leChRO-seq and RNA-seq) and that might affect the down-stream analysis and results. Can the authors comment on that?

Second, in the section 'Pachytene transcriptional burst activates genes involved in meiosis, transcription, and spermatogenesis' authors can provide the data as supplementary information (i.e., supplementary tables).

Regarding the DSB analysis, the authors do not find correlation between nascent transcription and the location of SPO11 oligos and PRDM9 sites. The authors might extend this analysis to PRDM9-no dependent DSB sites.

Taking strong consideration of the comments and suggestions from our three reviewers, we have now made substantial changes to our manuscript. We were happy to see that reviewers were unanimously excited about the hypotheses, approaches, and datasets presented in our original manuscript. For example, our reviewers noted that “The manuscript addresses an interesting and important topic...The experimental design is excellent, and the analysis of genomics data is well done” and that our data and analyses “leads to a much clearer model of how pachytene transcription burst works.”

We are thankful to the reviewers for the many highly constructive comments that have contributed significantly to improving our revised manuscript. In particular, we fully agree with points made about a lack of direct evidence for BRDT-dependent pause release in meiotic transcription and questions regarding our computational methods. In response, we have made several additions to the experimental design, content, analysis, and writing of our results. We believe that we have fully addressed the reviewers’ comments in the accompanying revision.

Changes of particular note include:

1. We more directly tested the hypothesis that BRDT and other bromodomain proteins are essential for Pol II pause release during the transition to pachynema by blocking BRDT using the small molecule, JQ1. This analysis, described in a new section of the main text, shows that blocking BRDT leads to an accumulation of paused Pol II (Ser5p) and a loss of transcriptionally active Pol II (Ser2p) in pachynema. These results are in agreement with our genomics analysis, indicating that BRDT is essential in pause release.
2. We have significantly improved the description of our analysis rationale and methods, including our description of A-MYB enrichment in ATAC-seq peaks and FACS sorting of prophase I cells, to provide critical clarifications to readers.
3. We have performed new analyses to analyze the pausing index on autosomes and sex chromosomes separately.

Please find below a point-by-point breakdown addressing the reviewers’ comments.

REVIEWER COMMENTS:

Reviewer #1 (Remarks to the Author):

The manuscript describes a detailed analysis of changes in transcription that take place during meiosis in mice. The manuscript explores the contribution of transcription initiation, promoter-proximal pausing, and RNA stability to these changes. The authors then examine the location of double-stranded breaks and their possible overlap with active enhancers and promoters. From these analyses, the authors conclude that, although DSBs are located in regions containing active histone modifications, their location is not transcribed and therefore, DSB do not overlap with transcribed genes.

The manuscript addresses an interesting and important topic. The experimental design is excellent, and the analysis of genomics data is well done. The manuscript is also logically written and easy to follow. I have no major concerns and I think the manuscript is appropriate for publication in Nat. Comm. The following are some comments the authors may want to consider:

Major Comments

1. ATAC-seq measures chromatin accessibility due to the presence of a bound transcription factor at the summit of the ATAC-seq peaks. The authors determined TFs responsible for transcription activation in pachynema by using dREG to identify eRNAs in leChRO-seq data. This seems like a less accurate method than using ATAC-seq peaks. Is A-MYB identified as the top TF when using ATAC-seq data? The authors find that the majority of A-MYB binding sites show ATAC-seq signal. Is the reverse also true, that most ATAC-seq peak summits contain A-MYB binding motifs? Do these peaks contain motifs for CREB and RFX2?

Response: The revised manuscript now includes the analysis suggested by this reviewer. Specifically, we include a new analysis in which we identify motifs enriched in ATAC-seq peaks that change during the transition to pachynema. This new analysis does indeed identify A-MYB, CREB, and RFX2 motifs in ATAC-seq peaks, as we have observed for dREG. These results are illustrated in the revised **Fig. S6**. This new analysis using the ATAC-seq data strongly supports the original conclusion from the dREG and leChRO-seq analyses.

It should be noted that the ATAC-seq method detects multiple different types of accessible chromatin, including promoters and enhancers, but also including insulator binding that serve a role in chromatin organization but not necessarily in transcription. Thus, not all ATAC-seq peak summits would be expected to contain motifs bound by transcription factors with a role in transcriptional activation. For reference, please see: **Klemm, S. L., Shipony, Z., & Greenleaf, W. J. (2019). Chromatin accessibility and the regulatory epigenome. Nature Reviews Genetics, 20(4), 207-220.**

For this reason, our view is that dREG is actually a more reliable source of active enhancer and promoter elements than ATAC-seq. To clarify this point in the revised manuscript, we have edited the text to read: "We identified the location of promoters and enhancers, collectively called transcriptional regulatory elements (TREs), during prophase I by using dREG to identify enhancer and promoter RNAs in leChRO-seq data from each prophase I stage... Similar motifs were obtained when using ATAC-seq data (**Fig. S6a**), but the order was different which reflects ATAC-seq marking additional types of functional elements, not all of which have a direct role in transcriptional regulation. Therefore, the discussion and analysis below is focused on dREG elements." For reference, please see: **Wang, Z., Chu, T., Choate, L. A., & Danko, C. G. (2019). Identification of regulatory elements from nascent transcription using dREG. Genome research, 29(2), 293-303.**

Minor comments

1. Line before last in the abstract. Perhaps "mechanisms" instead of "the mechanism" would be more appropriate in case there are other mechanisms.

Response: The recommended change has been made in the abstract.

2. Page 2 “RNAs transcribed during prophase I may also be stored until they are needed to facilitate events during transcriptionally inert stages later in spermiogenesis”. Could these RNAs be made in round spermatids?

Response: Previous studies have demonstrated that a large number of post-meiotic transcripts are expressed during prophase I. For reference, please see: Geisinger, A., Rodríguez-Casuriaga, R., & Benavente, R. (2021). Transcriptomics of meiosis in the male mouse. *Frontiers in Cell and Developmental Biology*, 9, 626020. However, in order to provide clarification that some RNAs are made in round spermatids, we added the following statement to the text: “In addition, several genes that are critical for spermiogenesis are expressed during prophase I, while others, such as the Y-linked *Zfy1* and *Zfy2* genes, must be transcriptionally repressed until they are needed to facilitate events during transcriptionally inert stages later in spermiogenesis.”

3. Bottom of page 2 and top of page 3 “a central question is how spermatocytes are able to balance the separate, and arguably opposing, tasks of chromatin decondensation to facilitate transcription and chromosome condensation to aid the defining events in prophase I”. Authors should consider whether the level of condensation (formation of large loops?) that results in the formation of prophase chromosomes is the same as the level of condensation that interferes with transcription (nucleosome positioning?).

Response: The reviewers point is well taken: It is largely unknown whether the level of chromosome condensation required for synapsis interferes with transcription and nucleosome positioning, as described here: Zuo, W., Chen, G., Gao, Z., Li, S., Chen, Y., Huang, C., ... & Bian, Q. (2021). Stage-resolved Hi-C analyses reveal meiotic chromosome organizational features influencing homolog alignment. *Nature communications*, 12(1), 1-20. One of the goals of our study is to determine how chromatin accessibility, nucleosome positioning, and transcription change during the defining events in prophase I. To provide clarification of our goal, we added the following text to the manuscript: “Currently, it remains unknown whether meiotic chromosome axis shortening, the formation of large chromatin loops, and the 3D genome reorganization of prophase I cells prohibit transcription and nucleosome positioning. Therefore, a central question is how spermatocytes are able to balance the distinct, and perhaps opposing, tasks of chromatin decondensation to facilitate transcription and meiotic chromosome axis formation to aid the defining events in prophase I.”

4. Figure 1. Perhaps it would be more intuitive to have Ser2p below Ser5p in this figure.

Response: The recommended change has been made to Figure 1.

5. Page 10. “Thus, we conclude that gene expression profiles for leptonema/zygonema, pachynema, and diplonema have notably different transcription programs”. Please explain how this conclusion was reached. It seems from the text that it was based on just a few genes. Alternatively, tone down the statement or delete the sentence.

Response: This sentence has been deleted from the manuscript.

6. Please refrain from referring to covalent histone modifications as “markers”.

Response: We have made the recommended correction throughout the entirety of the text.

7. Page 21, bottom. The authors seem to imply that DSBs do not overlap with ATAC-seq peaks. Is this correct? Please explain more explicitly in the text.

Response: We have provided additional clarification and explained our results more explicitly in the text: “These data suggest that, although DSBs are indeed located in regions of accessible chromatin, they are statistically less likely to occur at transcriptionally active regulatory elements.”

Reviewer #2 (Remarks to the Author):

In this manuscript, Alexander et al. generated time-series transcription-related omics data (mRNA-seq, leChRO-seq, and ATAC-seq) throughout the mouse meiosis prophase I to ask how spermatocytes achieve pachytene transcription burst and keep a balance between meiotic recombination and transcriptional activation through integrated analysis. The authors analyzed the average density of paused and elongated RNA polymerase in different stages through immunofluorescence staining and inferred that pachytene transcription burst is driven by released RNA pol II paused in leptotene and zygotene. Radioactive nuclear run-on and leChRO-seq data supported this in quantitative analyses. Then they identified the candidate TFs regulated the RNA pol II release via motif analysis using de novo dREG peaks found in pachytene and focused on MYBL1 and BRDT. By re-analysis of the previously reported MYBL1 and BRDT ChIP-seq data, the authors proposed a model of MYBL1-dependent recruitment of BRDT to highly paused genes in pachytene to release the paused RNA pol II. Finally, they compared transcribed regions and DSB hotspots and found these two regions are mutually exclusive on the genome despite with similar chromatin environments.

The omics data are of good quality, and the analysis was performed systematically and logically. The immunofluorescence staining of RNA pol II and leChRO-seq supports that the essence of pachytene transcriptional activation is releasing of paused Pol II. The occupancy sites of MYBL1 and BRDT showed a strong correlation with the pachytene-upregulated genes. Their targets showed a reduced pausing index in pachytene compared to leptotene/zygotene. This association leads to a much clearer model of how pachytene transcription burst works. However, I am concerned about the main conclusion that meiotic gene transcription programs are mediated by A-MYB and BRDT-dependent RNA polymerase II pause release because no data from knock-out mice were involved in this work. The leChRO-seq in *Brdt*^{-/-} pachytene spermatocytes may provide direct evidence for this hypothesis model.

Response: We have not yet been able to conduct leChRO-seq in *Brdt*^{-null} testes because of the severe paucity of spermatogenic cells. However, to address the reviewer's concern that we lack direct evidence for BRDT-dependent pause release in meiotic transcription, we instead used a pharmacological inhibition strategy to prevent BRDT function by injecting JQ1, a small molecule that blocks bromodomain protein function, in mice. In line with our hypothesis concerning BRDT function in Pol II pause release, we found that JQ1 treatment results in altered Pol II pause-release in prophase I spermatocytes. To demonstrate this, we have added a new figure (Figure 5) and section in the manuscript describing these data:

“To test our model for the involvement of BRDT in Pol II pause release in pachynema, we took a pharmacological approach to alter BRDT activity and explore pause release dynamics. We utilized the BET protein inhibitor thienodiazepine (+)-JQ1 (hereafter referred to as JQ1) to prevent BRDT function in prophase I cells. JQ1 inhibits the acetyl-lysine binding module of the BD1 domain of BRDT with high ligand efficiency¹⁸. JQ1 has been shown previously to inhibit BRDT function in spermatocytes, and this drug exhibits high testicular bioavailability without affecting hormone levels in male mice¹⁸. We injected 7-week-old male mice intraperitoneally (i.p.) with either JQ1 or vehicle solutions daily for 3 weeks (**Fig. 5a**). We performed immunofluorescence using antibodies recognizing Ser5P (paused Pol II) and Ser2P (elongating Pol II) to analyze meiotic chromosome spreads obtained from mice after 3 weeks of JQ1 or vehicle control (**Fig. 5 b and c**). We found significantly higher levels of Ser5P immunofluorescence signal in pachytene and diplotene spermatocytes from JQ1 treated mice when compared to control spermatocytes (*****p*-value < 0.0001; **Fig. 5 b and d**). By contrast, total Ser2P signal was significantly lower for JQ1 treated spermatocytes in pachynema and diplonema when compared to the vehicle treatment (*****p*-value < 0.0001; **Fig. 5 c and e**). Together, these data indicate that small-molecule inhibition of BRDT results in reduced levels of Pol II pause release in prophase I spermatocytes, leading to an accumulation of paused Pol II and a depletion of elongating polymerase.”

Specific comments:

1. In Figure 1, I noticed distinct signals of Ser5P near the sex chromosomes (1C), where no Pol II (1A) and Ser2P (1B) signal was detected. This indicates Pol II is paused at sex chromosomes which is opposite to autosomes in pachytene and diplotene. It is quite interesting to discuss this phenomenon from the perspective of Pol-II.

Response: We agree with the reviewer that finding paused Pol II on the sex chromosomes would be a very interesting finding. To address this comment, we carefully examined our IF images, including both those in the main figures and the many additional images that we took during our study. We did not detect any systematic enrichment of Ser5P signal near the sex chromosomes in pachytene cells. Additionally, looking back at our genomic data also did not identify any clear signal for large amounts of paused Pol II on the sex chromosomes. To provide additional clarification to readers, we have marked the location of the sex chromosomes in the revised Figure 1 (see the dotted ellipses in the revised figure).

2. The sex chromosome should be treated separately when performing transcription-related omic analysis due to MSCI. Are there any peaks of leChRO-seq at chrX in pachytene and diplotene? If so, how does the pausing index of chrX genes change at pachytene and diplotene compared to leptotene/zygotene?

Response: We agree fully with this comment. We detected only small amounts of leChRO-seq signal on the X and Y chromosomes. In the revised manuscript, we have performed the analysis suggested by the reviewer by examining the pausing index on the autosomes and sex chromosomes separately. These data show the same pattern on both autosomes and sex chromosomes (see the revised Figure 2 and Figure S2), in which pausing index of genes on the autosomes and sex chromosomes peaks in leptotene/zygotene and decrease during the successive stages of prophase I. These results show that our pausing index analyses of the autosomes are not driven by gene annotations located on the sex chromosomes.

3. I am confused about the negative values in leChRO-seq track in Fig 2C & Fig S3F. Authors should make a clearer description in figure legend.

Response: We have made the suggested edits to our figure legends: “Positive values represent the plus strand and negative values represent the minus strand.”

4. Some figure legends are not matched to corresponding figures (e.g. Fig S2, Fig. 4).

Response: Thank you for noticing this mistake. All figure legends have been updated.

5. In “A-MYB coordinates the pachytene transcriptional burst” part, the description of Fig S3A is not consistent with Fig. S3A

Response: Thank you for catching this. We moved the figure citation to the correct spot in the sentence.

6. In Fig 3B & C, the global cluster pattern should be provided.

Response: In order to show the global cluster pattern for Fig. 3b & c, we added a dendrogram showing the global hierarchical clustering pattern of genes in Figure S3.

7. The description of “A-MYB ChIP-seq data from pachytene spermatocytes showed a notable overlap (28%) with regulatory elements having MYB binding motifs” is quite confusing. Do authors compare A-MYB binding sites versus dREG with MYB motifs?

Response: We clarified the text and added “We next compared A-MYB ChIP-seq peaks to dREG peaks containing the binding motif for A-MYB. We found that A-MYB binding sites in pachytene

spermatocytes showed a notable overlap (28%) with regulatory elements having MYB binding motifs (Chi-squared test with Yates' correction, *** p -value < 0.0001)

8. In Fig. 5C & D, the union symbol may be an intersect symbol according to the description in the figure legends.

Response: We made the suggested correction to Fig. 5c & d (now Fig. 6c & d).

9. In Fig 6E, why only 5% of dREGs are overlapped with the ATAC-seq peaks, considering that transcribed regions should be accessible.

Response: Our thanks to the reviewer for catching this. There were two confusing points with this figure that we have made our best effort to resolve in the updated manuscript. First, Fig. 6E (now Fig. 7e) shows the fraction of all ATAC-seq peaks that are also transcribed. Second, we have caught an error in the data used for our analysis in the original figure panel. After correcting this error, the revised Fig. 7e shows that ~20% of all ATAC-seq accessible sites are also transcribed and identified using dREG. We know from other work that not all chromatin accessible regions are transcribed enhancers or promoters - additional cellular processes can make a position of the genome accessible to Tn5 (or DNase-I), for example CTCF binding in insulators. The number obtained in our revised analysis (~20%) is on roughly the same order of magnitude as detected in previous analyses (see: **Danko et. al. Nature Methods 2015; Wang, Chivu, et. al. Nature Genetics 2022**) and is extremely highly enriched.

10. In Fig S5, why overlapped regions of PRDM9 are less than DMC1, considering that more PRDM9 occupancy sites are observed than DMC1?

Response: We agree with this comment and understand how these results can be intuitively confusing. We note that the patterns of overlapped regions of transcribed ATAC-seq peaks with PRDM9 and DMC1 are similar. The absolute overlap with DMC1 may be greater for two reasons: (1) We used PRDM9 motifs located within SPO11 hotspots with a stringent match score cutoff to remove false positives as input for our analyses; and (2) In our dataset, there were more DMC1 ChIP-seq peaks identified than PRDM9 binding motifs. Therefore, the pattern we have identified is not particularly surprising. To clarify this, we added the following to the figure legend for Figure S6: "15,379 total PRDM9 binding motifs and 30,106 DMC1 ChIP-seq peaks were identified previously and used as input for panels A and B, respectively."

11. In methods, more information about which genes were used for PCA. All genes or genes with filtration?

Response: We clarified our PCA analysis in the methods: "Sequencing reads from each genomics assay were counted in gene bodies of all protein-coding genes that were shown to be expressed from the leChRO-seq data and used as input for the PCA analysis."

12. If loss of BRDT, the genes with a reduced pausing index in pachytene compared to leptotene/zygotene may have a tendency to be downregulated. The RNA-seq data of 20 dpp Brdt^{-/-} testis in GEO database under GSE39909 may be helpful.

Response: We thank the reviewer for providing this resource. However, the RNA-seq data to which the reviewer is referring to was not obtained from adult null males. Since steady-state pachynema does not occur at 20 dpp, these data will not pair with our genomic studies of 10-week-old males.

Reviewer #3 (Remarks to the Author):

In this work, Alexander and collaborators report a genomic profiling of Pol II occupancy in mouse primary spermatocytes. To do so, the authors combine leChRO-sequencing with RNA-seq and ATAC-seq in flow sorted enriched populations, distinguishing between leptotene/zygotene, pachynema and diplonema

spermatocytes. First the authors report an increased transcriptional activity in pachytene cells, suggesting that this is driven by the release of Pol II that has been previously paused in leptotene/zygotene. Following this finding, the authors seek to find which transcriptional factors might be driving this phenomenon, finding a high correlation to A-MYB binding sites. Not surprisingly, the authors do not find correlation between the location of DSBs and transcription.

The paper is well written, and I enjoyed going through it. I find the authors provide compelling evidence of the presence of paused Pol II at TSSs of highly expressed genes in early prophase I. I have though a couple of comments that might need further clarification.

First, it will be helpful to provide a more detailed description of the isolation of mouse spermatocytes by FACS. This can be added to the supplementary material, including a more detailed description of the methodology and flow cytometry plots showing different populations isolated.

Response: The revised manuscript now provides a more detailed description of the FACS procedure. To better explain our cell sorting strategy, we have also added a supplementary figure showing an example of the gating strategy used for isolation of prophase I substages by FACS in Figure S9.

I am bringing this up because it seems that in the PCA plots from Fig 2, the PC2 is not able to set completely apart pachytene from diplotene cells (leChRO-seq and RNA-seq) and that might affect the down-stream analysis and results. Can the authors comment on that?

Response: The data do indeed show that pachytene and diplotene cells have fairly similar transcription, mRNA, and chromatin accessibility profiles. However, we do not believe that an incomplete separation on PC2 will have a negative impact on our analysis for several reasons: First, pachytene and diplotene cells are well separated on PC1, which actually accounts for the majority of the variation in our FACS-sorted samples, which were used as input for ATAC-seq library preparation (PC1 = 95%). Second, as noted above, we think the similarity between pachytene and diplotene cells reflect the more similar biology of these two stages, rather than incomplete cell separation or other confounding factors. We note that we have extensively analyzed purity of all samples from STAPUT- and FACS-sorted cells used as input for our genomics data by staining against proteins defining prophase I substages, specifically SYCP3 and γH2AX, and evaluating the prophase I substage of at least 100 cells/ sample.

Second, in the section ‘Pachytene transcriptional burst activates genes involved in meiosis, transcription, and spermatogenesis’ authors can provide the data as supplementary information (i.e., supplementary tables).

Response: We have included an excel file with the names of genes for each cluster presented in Fig. 3b-c as supplementary information with our resubmission.

Regarding the DSB analysis, the authors do not find correlation between nascent transcription and the location of SPO11 oligos and PRDM9 sites. The authors might extend this analysis to PRDM9-independent DSB sites.

Response: We have performed the analysis suggested by the reviewer in our revised manuscript. In this new analysis, we analyzed the landscape of chromatin accessibility and transcription around PRDM9-independent DSB sites, and added the data to Figure S7b-c. We also added the following text to report our findings to readers: “We also note that PRDM9-independent DSBs show a reduction in chromatin accessibility levels from leptotene/zygotene to pachynema (Fig. S7b). However, whereas promoters show a strong signal for transcription that peaks in pachynema (Fig. 7c), both PRDM9-dependent and independent DSBs show no evidence of transcriptional activity at any point during prophase I (Fig. 7d; Fig. S7c).”

REVIEWERS' COMMENTS

Reviewer #1 (Remarks to the Author):

The authors have addressed all my initial concerns, which were quite minor. This is an excellent manuscript on an interesting topic describing significant results in the field. The manuscript is appropriate for publication in Nat Comm

Reviewer #2 (Remarks to the Author):

I have read the author's reply and the revised manuscript. I thought the authors had answered my questions and concerns, and I was basically satisfied with their answers.

However, I don't understand why the Pausing index on sex chromosomes is completely consistent with that on autosomes, because in prophase I, the change trends of gene expression on the sex chromosomes and autosomes are obviously inconsistent or even contrary. In particular, a couple of recent scRNA-seq analyses did not detect MSCI escapees during pachytene and diplotene stages in male mice (Jung et al., 2019; Shami et al., 2020). This could suggest that Pausing index cannot fully reflect gene expression, especially for the genes on the sex chromosomes. I suggest the authors to add a paragraph talking about this in discussion section before the manuscript formally accepted.

Reviewer #3 (Remarks to the Author):

Authors have responded satisfactorily to all my previous comments. Congratulations on a beautiful paper.

Manuscript: A-MYB and BRDT-dependent RNA Polymerase II pause release orchestrates transcriptional regulation in mammalian meiosis

Response to the Reviewer

Reviewer #2 (Remarks to the Author):

I have read the author's reply and the revised manuscript. I thought the authors had answered my questions and concerns, and I was basically satisfied with their answers.

However, I don't understand why the Pausing index on sex chromosomes is completely consistent with that on autosomes, because in prophase I, the change trends of gene expression on the sex chromosomes and autosomes are obviously inconsistent or even contrary. In particular, a couple of recent scRNA-seq analyses did not detect MSCI escapees during pachytene and diplotene stages in male mice (Jung et al., 2019; Shami et al., 2020). This could suggest that Pausing index cannot fully reflect gene expression, especially for the genes on the sex chromosomes. I suggest the authors to add a paragraph talking about this in discussion section before the manuscript formally accepted.

We made the requested change in the manuscript. We added a paragraph to the discussion section regarding our interpretation of the pausing index changes that occur on the sex chromosomes during prophase I progression.

We added the following text to the manuscript:

“In mammalian meiosis, autosomes that fail to undergo synapsis and/or recombination are subject to a surveillance mechanism termed meiotic silencing of unsynapsed chromatin (MSUC). MSUC deprives germ cells of transcripts via a megabase-scale chromatin remodeling process. Meiotic silencing events are also associated with asynapsed regions of the X and Y chromosomes, which undergo synapsis and recombination only at the ~1 Mb pseudoautosomal region. However, unlike MSUC, meiotic sex chromosome inactivation (MSCI) is an obligatory event that arises as spermatocytes enter pachynema and is concurrent with the compartmentalization of the XY chromosomes into a heterochromatin rich nuclear subdomain known as the sex body. Disruption of MSCI induces mid-pachytene disruption of meiosis, leading to loss of germ cells and infertility. It has remained unclear whether XY chromatin is permissive to Pol II recruitment and initiation at select loci. Here, we note that the sex chromosomes show a similar reduction in Pol II pausing index to that observed on the autosomes in pachynema, but without a concomitant increase in transcriptional activity. In the current study, the use of a highly sensitive assay to detect nascent RNA (leChRO-seq) allowed us to capture this pattern of pause release on the X and Y chromosomes, which would not be observable through bulk RNA-seq alone. The decrease in Pol II pausing indices on the sex chromosomes can be interpreted in several ways. First, assuming that A-MYB is capable of binding to XY chromatin in pachynema, induction of pause release on the sex chromosomes would be wholly possible, just as it would be on the autosomes. Second, mRNA expression of sex-linked loci may be low due to the formation of XY heterochromatin preventing productive elongation of Pol II or other mechanisms inducing rapid termination of paused Pol II on the sex chromosomes. Together, these data indicate that a reduction in Pol II pausing indices at XY genes in pachynema is not coupled with transcriptional activation as was observed on the autosomes. The precise nature of Pol II pause release regulation on the sex chromosomes will be a subject of investigation in our future studies.”